# Molecular simulations of enzymatic phosphorylation of disordered proteins and their condensates

Emanuele Zippo [1,2], Dorothee Dormann [2,3], Thomas Speck[4] & Lukas S. Stelzl [2,3,5] ✉

Condensation and aggregation of disordered proteins in cellular non-equilibrium environments are shaped decisively by enzymes. Enzymes called kinases phosphorylate proteins, consuming the chemical fuel ATP. Protein phosphorylation by kinases such as Casein kinase 1 delta (CK1$\delta$) determines the interactions of neurodegeneration-linked proteins such as TDP-43. Hyperphosphorylation of TDP-43 by CK1$\delta$ may be a cytoprotective mechanism for neurons, but how CK1$\delta$ interacts with protein condensates is not known. Molecular dynamics simulations hold the promise to resolve how kinases interact with disordered proteins and their condensates, and how this shapes the phosphorylation dynamics. In practice, it is difficult to verify whether implementations of chemical-fuel driven coarse-grained simulations are thermodynamically consistent, which we address by a generally applicable and automatic Markov state modeling approach. In this work, we thus elucidate with coarse-grained simulations, drivers of how TDP-43 is phosphorylated by CK1$\delta$ and how this leads to the dissolution of TDP-43 condensates upon hyperphosphorylation.

Biological systems operate far from equilibrium[1]. The functionalities of cells and of their organelles and compartments are possible only through their precise self-organization, driven by a continuous injection of energy from the external environment[2]. In the cell, chemical energy is stored, e.g., in the form of adenosine triphosphate (ATP) molecules[3]. This energy is then used to synthesize and degrade molecules. On time scales shorter than physiological changes, microscopic rates are approximately constant and the system enters a non-equilibrium steady state (NESS)[3–5].

An important contributor to cellular self-organization are membrane-less protein condensates, and dysregulation of protein condensates is linked to the development of neurodegenerative diseases[6,7]. Biological polymers such as proteins can phase-separate, giving rise to biomolecular condensates[6]. Phase-separated proteins can concentrate molecules, including ATP[8,9], while excluding other

molecules from these membrane-less compartments. Consequently, these phase-separated protein condensates can organize biochemical processes in time and space, which is analogous to the compartmentalization provided by lipid membranes[6,10]. However, condensates can age and become less liquid-like, and in the process, lose their biochemical functionalities[11]. Condensates of proteins can also age into solid aggregates, which are believed to contribute to neuronal dysfunction and neurodegeneration[7,12–14]. Many condensation and aggregation-prone proteins are intrinsically disordered proteins (IDPs) or feature extended disordered regions (IDRs) lacking a well-defined 3D structure. A paradigmatic example is TAR DNA-binding protein 43 (TDP-43), which, in addition to folded domains, has a disordered low complexity domain (LCD, residues 261-414) at its C-terminus. TDP-43 condensates are functionally important[15–17]. The LCD is critical for the functional roles of TDP-43 in, e.g., splicing and 3' poly-adenylation[15,16,18].

[1]Institute of Physics, Johannes Gutenberg University Mainz, Mainz, Germany. [2]Institute of Molecular Physiology, Johannes Gutenberg University Mainz, Mainz, Germany. [3]Institute of Molecular Biology (IMB), Mainz, Germany. [4]Institute for Theoretical Physics IV, University of Stuttgart, Stuttgart, Germany. [5]KOMET1, Institute of Physics, Johannes Gutenberg University Mainz, Mainz, Germany. ✉e-mail: lstelzl@uni-mainz.de

A conserved helix (residues 319–341) within the LCD has been shown to be important for phase separation[19,20] and the physiological roles of TDP-43 in RNA processing[15,18]. Besides the helical structure in the LCD, its enrichment in aromatic and aliphatic residues, including methionine, drive its phase separation[21–23]. In diseased neurons, TDP-43 loses its nuclear localization and forms aggregates in the cytoplasm[17,20]. TDP-43 aggregates are found in patients with amyotrophic lateral sclerosis (ALS)[24], and frontotemporal dementia[25], but also in many patients with Alzheimer's disease[26].

Proteins within condensates can also undergo chemical reactions themselves[2], driving the system out of equilibrium by dissipating a biochemical fuel, such as ATP. The modifications of those proteins by the addition of chemical groups, such as phosphate groups, are referred to as post-translational modifications (PTMs). PTMs are frequently found on IDRs and IDPs[27]. PTMs can drastically change the properties of individual proteins[28] and collectively of condensates[29], enhancing[13,30] or suppressing the condensation and aggregation of IDPs[31,32]. For instance, it has been shown that chemical reactions can stabilize the size of liquid droplets by suppressing Ostwald ripening[33,34].

To connect these advances in the understanding of active processes in condensates to the biological roles of proteins, it will be important to elucidate how ATP-driven phosphorylation shapes the interactions of IDRs of neurodegeneration-linked proteins such as TDP-43. Phosphorylation of the TDP-43 C-terminal residues Ser 379, Ser 403, Ser 404, Ser 409, and Ser 410 in protein aggregates from patient samples is associated with neurodegenerative disease[35]. Ser 409/Ser 410 phosphorylation has been established as a hallmark of TDP-43 pathology[35] and can be detected, together with Ser 403/Ser 404 and Ser 379, by phospho-specific antibodies[36–38]. It is not clear whether TDP-43 aggregates or condensates are phosphorylated, or TDP-43 is phosphorylated in solution. TDP-43 phosphorylation may also disrupt the cognate interactions of TDP-43, which underpin its physiological functions[16,17]. TDP-43 LCD hyperphosphorylation has been shown to suppress TDP-43 condensation and aggregation, which suggests that phosphorylation may be cyto-protective rather than driving pathogenesis[39]. TDP-43 is phosphorylated by Casein kinase 1δ (CK1δ) among other kinases[40]. Phosphorylation of one Ser can prime another Ser residue for phosphorylation[41]. Enzymes can add PTMs to IDPs in dilute solution, but enzymatic addition of PTMs may also occur in protein condensates. Recently, it was shown that the enzymatic phosphorylation of Tau was accelerated by protein condensation[42]. How the enzymatic phosphorylation of TDP-43 is modulated in dilute solution and how it is affected by protein condensates is not known. The disordered tail of CK1δ is auto-inhibitory[43,44], but how it inhibits TDP-43 phosphorylation is unclear on the molecular scale. IDRs of enzymes have multiple functions, such as auto-inhibition by binding to the active site. IDRs are involved in substrate binding, for instance, IDRs can speed up reactions via the fly-casting effect, where the IDR increases the search volume for the binding of partner proteins[45].

Molecular dynamics (MD) simulations can reveal important drivers of protein phase separation and condensation[39,46–53], but chemical reactions in condensates have been simulated only very recently[54,55]. Coarse-grained MD simulations capture the spontaneous condensation of hundreds or more proteins while maintaining enough chemical detail in the simulations to elucidate sequence-specific interactions of proteins[47,48,50,56,57]. However, most of these studies assume thermodynamic equilibrium, neglecting the dynamical changes in the properties of protein condensates[55], as well as the dissipation of chemical fuel due to biochemical reactions. Much progress has already been made in the simulations of mechanically-driven NESS, where external mechanical forces give rise to driven dynamics[58]. An important step was the construction of Markov state models[59] to better understand the effects of driving on the molecular scale[60]. Analogously, a biological chemically-driven NESS, such as molecular

motors, can be simulated by maintaining a chemical potential difference, i.e., by fixing the ATP to ADP concentration ratio[4,61]. Chemical reactions could in principle also be modeled via quantum mechanical approaches[62], but these are computationally very demanding, which can preclude their application to large-scale dynamics in complex biochemical systems. Recently, exciting progress has been made in integrating chemical reactions in molecular dynamics simulations via neural networks[63]. Even in the case of coarse-grained simulations, chemical reactions have been modeled through the use of reactive beads that can form bonds between molecules[54,55]. In many cases, one could model chemical reactions in a complex system by combining MD with a suitably chosen Monte Carlo (MC) step[55,61,64]. Arguably, the absence of a straightforward approach of validating the thermodynamic consistency of simulations of NESS has held back the widespread application of MD/MC approaches to biochemical reactions on the molecular scale.

Here, we show with coarse-grained molecular dynamics simulations the drivers of how TDP-43 is phosphorylated by the enzyme CK1δ, both in dilute solution and condensates. We demonstrate how to validate the thermodynamic consistency of simulations of enzymatic phosphorylation of proteins. We do so by constructing a Markov state model (MSM), which is a generally applicable approach, in the sense that it can be used to establish thermodynamic consistency for different simulation approaches, but also to investigate chemically-driven dynamics of biomolecules. Our coarse-grained simulations of enzymatic phosphorylation of TDP-43 show how the sequence-specific interactions of CK1δ with TDP-43 LCD affects the phosphorylation frequency of serine residues in the TDP-43 LCD in dilute solution and in condensates. In particular, the C-terminal domain (Ser 369 to Ser 410) is more phosphorylated than the N-terminus (Ser 266 to Ser 350), in agreement with experiments. Besides the sequence composition, i.e., the presence of charged and aromatic residues, and the sequence context, the distribution of Ser residues is an important determinant of phosphorylation patterns. The phosphorylation of one Ser can favor the phosphorylation of other Ser residues, in line with experiments, since the interactions between CK1δ and TDP-43 change after each reaction, enhancing further phosphorylations. Moreover, we study the role of the CK1δ IDR (residues 295–415) in phosphorylating TDP-43 both in the condensate and dilute regimes. CK1δ IDR interacts favorably with TDP-43 LCD, helping its recruitment of condensates. By binding to the condensates, CK1δ can phosphorylate TDP-43 in condensates and ultimately dissolve the condensates.

## Results

### Markov-state models of simulations of chemically-driven dynamics

Molecular dynamics (MD) simulations, together with a thermostat holding the temperature fixed, can be employed to sample from the canonical equilibrium distribution. However, introducing phosphorylation reactions in MD simulations generally injects energy into the system, thus breaking detailed balance and displacing the system away from thermal equilibrium. We simulate the action of the kinase CK1δ (residues 3–294) on the substrate protein TDP-43 LCD by combining one-bead-per-residue implicit-solvent MD (Supplementary Methods) with MC phosphorylation steps and validate the thermodynamic consistency of our simulations by making use of Markov state models (MSMs)[59]. We assume that only the serines (Ser) of TDP-43 LCD can be phosphorylated into phospho-serines (pSer). The phosphorylation reaction is the following:

$$\text{Ser} + \text{ATP} \rightleftharpoons \text{pSer} + \text{ADP} \qquad (1)$$

Whenever a Ser (or pSer) is in contact with the active site of the kinase during the MD simulation, we try to swap it through a Metropolis-like step[65] with a pSer (or the opposite) with acceptance

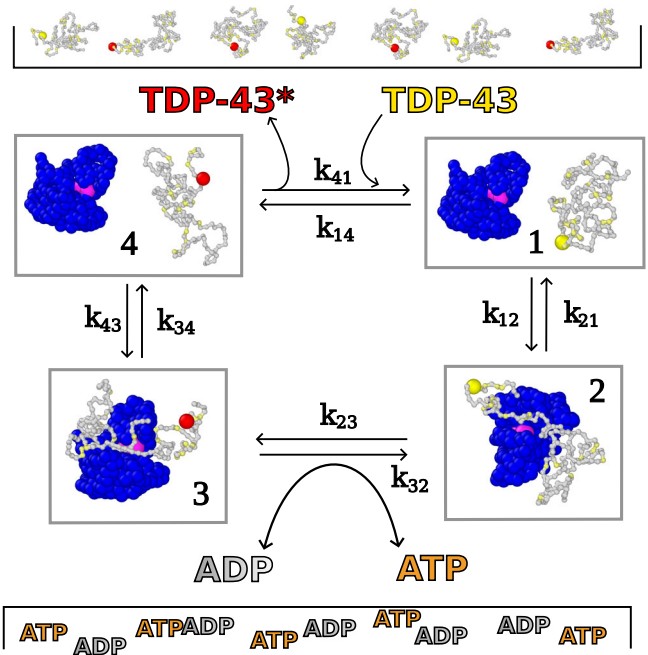

**Fig. 1 | Enzymatic phosphorylation cycle driven by ATP consumption.** In state **1**, TDP-43 (gray) is unphosphorylated and is not bound to the kinase CK1δ (blue, active site in pink). In state **2**, TDP-43 binds to CK1δ. In state **3** the reactive serine is phosphorylated by kinase, converting one ATP into one ADP. In state **4**, phosphorylated TDP-43 dissociates from CK1δ. Phosphorylated and unphosphorylated TDP-43 are supplied through reservoirs, and we consider exchanges between these reservoirs and our simulation box. Serines are colored in yellow, while phosphoserines in red.

probability given by

$$A(\text{Ser}, \text{pSer}) = \min\left(1, \exp(-\beta \Delta U_{\text{P}} - \beta \Delta \mu_{\text{P}})\right) \tag{2}$$

where $\beta = 1/(k_B T)$, $\Delta U_{\text{P}}$ is the difference between the potential energy of the configuration with pSer and that of the configuration with Ser, and $\Delta \mu_{\text{P}}$ is the chemical potential difference between the ATP and ADP molecules involved in the phosphorylation reaction in Equation (1) (Methods). ATP and ADP are modeled implicitly and are not explicitly simulated, with concentrations kept fixed and fully characterized through the choice of $\Delta \mu_{\text{P}}$. Indeed, biological reactions, such as the phosphorylation reaction, in living cells happen in open systems in which the concentrations of substrates, products, and the chemical fuel are kept approximately constant over relevant timescales by, e.g., metabolic processes[4]. A non-zero value of $\Delta \mu_{\text{P}}$ biases the chemical reaction, pushing the simulation away from thermodynamic equilibrium.

We validate the thermodynamic consistency of our simulations by showing that the dissipated heat in a phosphorylation cycle (referred to as $\Delta \mu_{\text{cycle}}$ in the following) is equal to the chemical potential difference $\Delta \mu_{\text{P}}$ in the phosphorylation step. The simplest example of a phosphorylation cycle that we can build is a system with one enzyme and one substrate protein in which only one residue is reactive. In order to get complete phosphorylation cycles, we assume the exchange between TDP-43 and phosphorylated TDP-43 happens when substrate and enzyme are far away from each other, without chemical driving and with equilibrium concentrations, through another MC step (Methods). This naturally happens in cells through the action of phosphatases that can catalyze a dephosphorylation reaction (pSer ⇌ Ser + P$_i$). In order to compute $\Delta \mu_{\text{cycle}}$, we employ a discretization of the MD trajectory in a 4 state MSM, as shown in Fig. 1. Assuming that our discretized trajectory stabilizes in a NESS, we can then compute the

transition rates $k_{ij}(\tau)$ from state $i$ to state $j$ and use the local detailed balance condition to compute the dissipated energy in one forward cycle $1 \to 2 \to 3 \to 4 \to 1$[3,5]:

$$\Delta \mu_{\text{cycle}} = -RT \ln\left(\frac{k_{12} k_{23} k_{34} k_{41}}{k_{14} k_{43} k_{32} k_{21}}\right). \tag{3}$$

It is interesting to observe that $\Delta \mu_{\text{cycle}}$ is exclusively determined by the ratio between forward and backward transition rates, while their absolute value is not relevant by itself. We computed the transition rates $k_{ij}$ from the transition probability matrix $T_{ij}(\tau)$, with $\tau$ the lag time, using a first-order approximation in $\tau$ (Supplementary Note 1). $T_{ij}(\tau)$ is estimated from the discretized trajectory through a non-reversible Maximum Likelihood estimator[66,67]. The transitions $1 \rightleftharpoons 2$, $3 \rightleftharpoons 4$ (the binding/unbinding of the enzyme to/from TDP-43 or phosphorylated TDP-43) and $4 \rightleftharpoons 1$ (the reservoir exchange step) are not driven by external energy sources, thus we expect the ratio of rates $k_{ij}/k_{ji}$ to be independent from $\Delta \mu_{\text{P}}$ and be determined by the equilibrium free energies of the states. By contrast, the phosphorylation MC step $2 \rightleftharpoons 3$ is driven by $\Delta \mu_{\text{P}}$, which creates a net probability current (flowing clockwise in the sketch of Fig. 1 if $\Delta \mu_{\text{P}} < 0$) that breaks the detailed balance condition[58]. We then expect $k_{23}/k_{32}$ to be determined not only by the equilibrium free energies, but also by $\Delta \mu_{\text{P}}$. For these reasons, we expect the probability flux to dissipate an amount of heat per cycle $\Delta \mu_{\text{cycle}}$ equal to the $\Delta \mu_{\text{P}}$ (Supplementary Note 2). We note that the absolute values of $k_{23}$ and $k_{32}$ depend also on the attempt rate of the MC step (Methods), which, though does not affect their ratio (Supplementary Fig. 1 and Supplementary Note 2).

The first step to discretize the MD/MC trajectory into a 4-state MSM is to distinguish between bound and unbound configurations. We achieved this by using a neural network called VAMPnet[68]. VAMPnet is able to map molecular coordinates to Markov states through a score function called the VAMP-2 score based on Koopman's theory. Finding the transformation of the input variables that maximizes the VAMP-2 score is equivalent to optimizing the Markovianity of the output states. In this way, we can easily distinguish between the two slowest processes, binding and unbinding, without arbitrarily choosing an a priori criterion of contact. As input for the neural network, we use the 154 distances between each residue of TDP-43 LCD and the active site of CK1δ, while as output, we ask for 2 states (ideally bound and unbound). As an example, we show in Fig. 2a how the distance between Ser 403 (the reactive residue) of TDP-43 LCD and the active site of CK1δ changes over the course of the trajectory for the simulation at $\Delta \mu_{\text{P}} = -5$ kJ mol$^{-1}$ (Supplementary Movie 1). We can see that the two states predicted by the neural network comprise bound configurations (when the distance between Ser and CK1δ active site is smaller) and unbound configurations (when the distance is larger).

By distinguishing between Ser and pSer along the trajectory, we coarse-grain the system dynamics into the 4 states sketched in Fig. 1. We report in Fig. 2b the resulting MSM discretized trajectory for the simulation in panel a. Complete cycles $1 \to 2 \to 3 \to 4 \to 1$ are highlighted in red. Every step of the Markov chain corresponds to $10^4$ MD steps, or 0.1 ns in simulation time. We then compute the transition probabilities $T_{ij}(\tau)$, as shown in Fig. 2c for the example trajectory. For all our simulations, we choose a lag time $\tau = 10$ Markov chain steps (Methods). We show in Fig. 2d the implied timescales for the example case of reactive Ser 403 and $\Delta \mu_{\text{P}} = -5$ kJ mol$^{-1}$. In the end, we estimate the reliability of the MSM by looking at the Chapman-Kolmogorov test (CK test)[59,69]. In all the validation simulations, the CK tests suggest good agreement between model and prediction for a wide range of lag times, as shown in Fig. 2e for the example case of reactive Ser 403 and $\Delta \mu_{\text{P}} = -5$ kJ mol$^{-1}$ (Supplementary Table 1, 2 and Supplementary Fig. 3 for complete data, Methods).

Finally, we compute the estimated dissipated heat per cycle $\Delta \mu_{\text{cycle}}$ from the transition rates $k_{ij}$ (Equation (3)) and plot them against

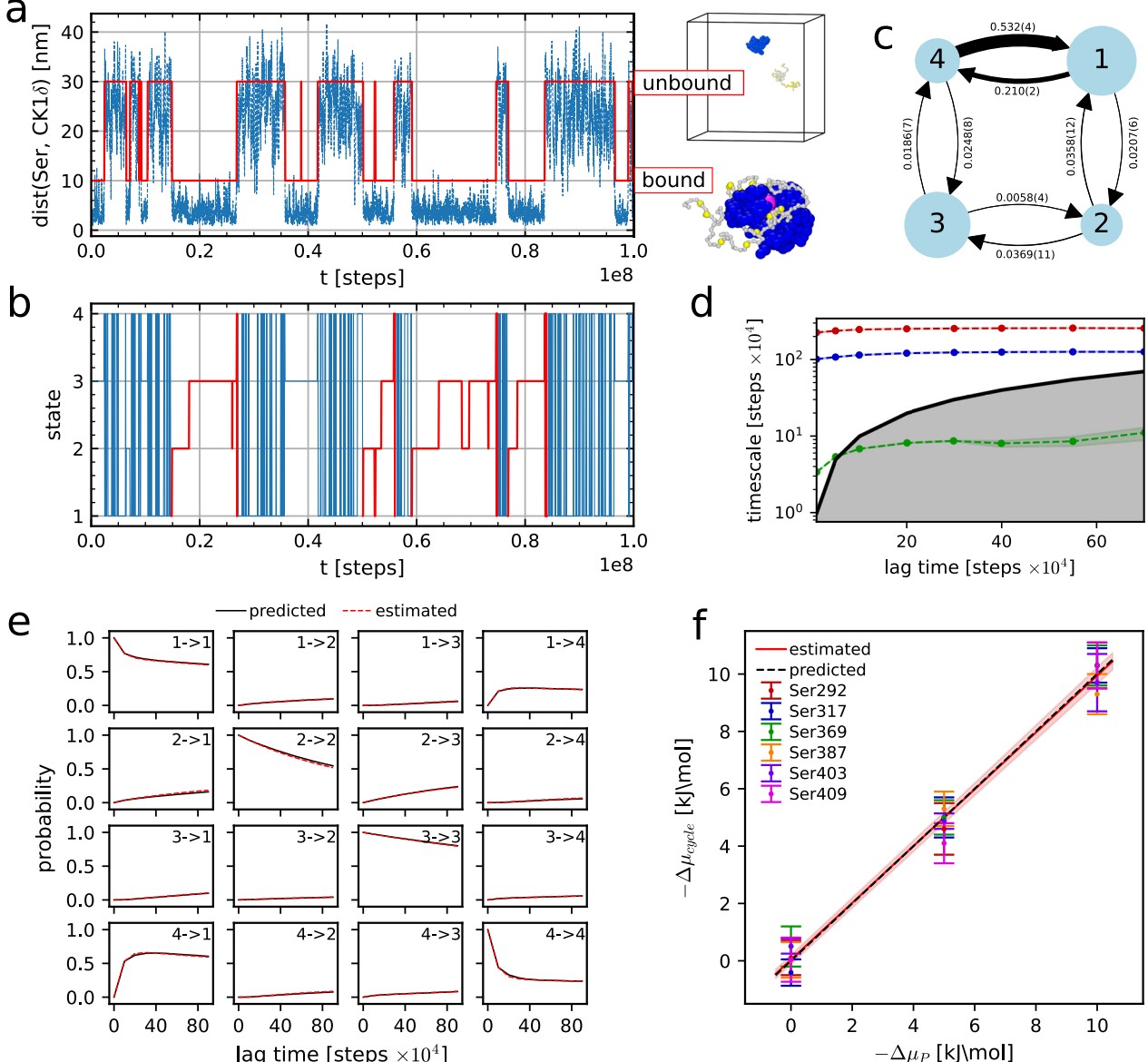

**Fig. 2 | Thermodynamic consistency of simulations with phosphorylation step using MSMs. a** Example of trajectory of the distance between Ser 403 of TDP-43 LCD and CK1$\delta$ active site from simulation at $\Delta\mu_P = -5$ kJ mol$^{-1}$. In red, the two output states of the neural network (bound and unbound) with respective illustrative examples on the right. **b** Example of discretized 4-state MSM trajectory related to the trajectory in **a**, we highlight complete cycles in red. **c** Sketch of the MSM with transition rates related to the trajectory in **a**. The size of the arrows is proportional to the absolute value of the corresponding transition rate, and the size of the circular node is proportional to the stationary distribution of the corresponding state. **d** Mean values (dots, connected with dashed line) +/− SEM (shaded area) of

implied timescales from the 3 eigenvalues of $T_{ij}(\tau)$ different from 1 (one for each color), estimated by bootstrapping 100 samples from the 100 μs long discretized trajectory shown in **b** at different lag times. **e** Example of Chapman-Kolmogorov test from the 4-state MSM trajectory shown in **b**. Shaded areas represent 95% confidence intervals from 100 bootstrapped samples. **f** We plot mean values +/− SEM of $\Delta\mu_{cycle}$ vs $\Delta\mu_P$, estimated by bootstrapping 30 samples from the 20 μs long discretized trajectories (100 μs for Ser 403 and $\Delta\mu_P = 0, -5$ kJ mol$^{-1}$). The red line with shaded area represents the best-fit line with propagated error from the parameter estimate. The predicted dashed line represents $\Delta\mu_{cycle} = \Delta\mu_P$. Raw data provided in the Source Data file.

the parameter $\Delta\mu_P$ of the phosphorylation step for different reactive Ser and $\Delta\mu_P$. Encouragingly, for all the six different phosphorylation sites, $\Delta\mu_{cycle}$ computed from Equation (3) matches the applied chemical potential $\Delta\mu_P$ (Fig. 2f). We also observe that the logarithm of the rate ratios $RT \log(k_{ij}/k_{ji})$ for the transitions $1 \rightleftharpoons 2$, $3 \rightleftharpoons 4$ and $4 \rightleftharpoons 1$ are independent from $\Delta\mu_P$, as these transitions are not subjected to chemical driving, while that for $2 \rightleftharpoons 3$ grows linearly with $\Delta\mu_P$, as expected for a transition driven by the dissipation of the chemical fuel (Supplementary Fig. 2). We checked the robustness of our conclusions, by determining $\Delta\mu_{cycle}$ from 1) a 3-state MSM and 2) using additional input distances for the VAMPnet neural network and find that we can still

verify the cycle relation from the simulations (Supplementary Note 3, Supplementary Fig. 3 and Fig. 4).

## C-terminus of TDP-43 is preferentially phosphorylated

Having established a model of chemically driven dynamics, we investigate how sequence context determines the phosphorylation of the disordered protein TDP-43 LCD by the enzyme CK1$\delta$[35,36,39]. In our simulations, we follow directly the dynamics of TDP-43 LCD and CK1$\delta$ folded domain (residues 3–294, Supplementary Fig. 5, 6) on the single molecule level (Fig. 3a). We run 100 simulations of TDP-43 LCD in presence of CK1$\delta$ and at physiological ATP/ADP ratio

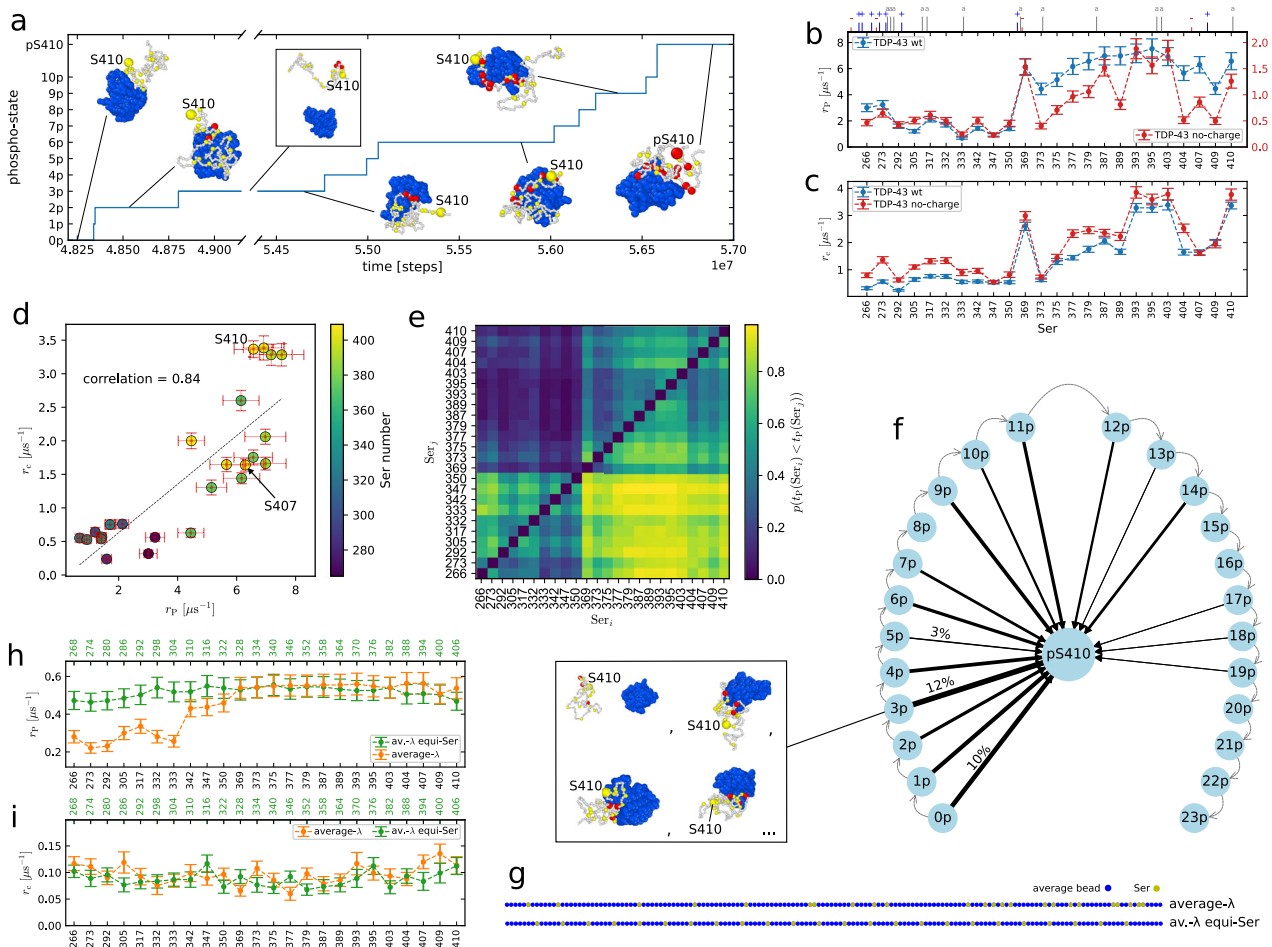

**Fig. 3 | Sequence dependence in phosphorylation dynamics of TDP-43.**
**a** Example trajectory discretized in phosphorylation-states relative to Ser 410. $n$p is the state with Ser 410 not phosphorylated and $n$ other pSer; pS410 refers to all the states with phosphorylated Ser 410. **b** Mean values +/− SEM of phosphorylation rates $r_P$ for every Ser of wild-type TDP-43 LCD (blue) and TDP-43 without charges (red, right y-axis) from 100 replicas. The ticks on top show the position of the charged (blue `+' positive, red `−' negative) and aromatic (gray `a') residues. **c** Mean values +/− SEM of contact rates $r_c$ between Ser residues of TDP-43 LCD (wild type in blue, no-charge sequence in red) and the active site of CK1$\delta$ in equilibrium simulations without phosphorylations from 30 replicas. **d** Correlation plot of contact frequency in equilibrium $r_c$ (mean +/− SEM from 30 replicas) and phosphorylation rate $r_P$ (mean +/− SEM from 100 replicas) for the wild-type TDP-43. **e** Probability $p(t(Ser_i) < t(Ser_j))$ of $Ser_i$ being phosphorylated before $Ser_j$, data from 100

trajectories. **f** Histogram of arrival orders for the phosphorylation of Ser 410. The thickness of the arrows represents the percentage of simulations in which Ser 410 was phosphorylated after $n$ other Ser residues (e.g., 12% of simulations go from state 3p to pS410, 0% from 20p). The gray dashed arrows from $n$p to $(n+1)$p illustrate the temporal succession of the states. Inset: Examples of 3p states. **g** Representation of the Ser position for averaged-interaction sequence with Ser at original positions (`average-$\lambda$') and for the averaged-interaction with equally-spaced Ser (`av.-$\lambda$ equi-Ser'). Ser residues beads in yellow and averaged interaction strength beads in blue. **h, i** Mean values +/− SEM of phosphorylation rates $r_P$ (**h**) from 100 replicas and contact rates $r_c$ (**i**) from 30 replicas for the averaged-interaction sequence with Ser at original positions (orange, `average-$\lambda$' in **g**) and for the averaged-interaction with equally-spaced Ser (green, top x-axis, `av.-$\lambda$ equi-Ser' in **g**). Raw data provided in the Source Data file.

($\Delta\mu_P = -48$ kJ mol$^{-1}$, Supplementary Note 4), which mimics in vitro kinase assays. In the simulations, unphosphorylated TDP-43 LCD will eventually encounter CK1$\delta$ and Ser residues will be phosphorylated by CK1$\delta$, as shown for an example simulation in Fig. 3a. In this simulation, TDP-43 LCD is initially phosphorylated in the C-terminal region. The kinase dissociates after two phosphorylation events and then binds again to the substrate. Multiple Ser residues in the C-terminus of TDP-43 LCD are phosphorylated, including Ser 410, which is phosphorylated after ten other residues.

In our simulations, Ser residues towards the C-terminus of TDP-43 LCD (Ser 369 to Ser 410) are more readily phosphorylated than Ser residues in the N-terminal region of the LCD (Ser 266 to Ser 350), with the phosphorylation rate $r_P$ on average roughly 3-4 times higher in the C-terminal segment than in the N-terminal segment (Fig. 3b). In mass spectrometry analysis of TDP-43 from ALS patient samples, the phosphorylated sites (the 12 residues Ser 373, Ser 375, Ser 379, Ser 387, Ser 389, Ser 393, Ser 395, Ser 403, Ser 404, Ser 407, Ser 409 and Ser 410[36,39])

and also Ser 369[37]) are mostly in the C-terminal region and, interestingly, they are among those with highest phosphorylation rate $r_P$ in our simulations. In particular, Ser 409/Ser 410 phosphorylation has long been established as a hallmark of TDP-43 pathology in disease[35,38]. This qualitative agreement with residue-level coarse-grained simulations tentatively suggests that sequence-specific interactions of TDP-43 LCD with the CK1$\delta$ could explain why these residues are frequently found phosphorylated in experiments and in patient samples.

The differences in the phosphorylation rates can be largely accounted for by how readily Ser residues engage in contacts with the CK1$\delta$ active site (Fig. 3c and Supplementary Fig. 5a, b). The phosphorylation rates per residue are strongly correlated with the frequency with which a residue is in contact with the active site in equilibrium MD simulations, with a sample Pearson correlation coefficient of 0.84 (Fig. 3d). In order to compare the phosphorylation rates with the frequency of making contacts at equilibrium, we performed MD simulations of the same system without phosphorylation MC

steps. To establish to what extent contacts predicts the relative phosphorylation rates, we consider a contact whenever all three distances to residues Asp 149, Phe 150, and Gly 151 close to the active site are less than 1 nm, in the same way as for the MC phosphorylation step. By contrast, the acceptance probability for the phosphorylation MC step for Ser residues once they are in contact is > 0.90 for the entire sequence and the variations in the acceptance probability of the phosphorylation step are not correlated with the variation in the phosphorylation rates (Supplementary Fig. 7). Ser residues in the C-terminal segment of the LCD, including Ser 369, Ser 393, Ser 395, Ser 403, and Ser 410, have the largest tendencies to form contacts, as tracked by $r_c$, which is the rate at which a residue forms contacts with the CK1$\delta$ active site (Fig. 3c). At the same time these residues have, within the statistical uncertainty, the fastest phosphorylation rates of the TDP-43 LCD (Fig. 3b). Ser residues in the N-terminal part of the LCD (Ser 266 to Ser 350) form fewer contacts than serines in the C-terminal segment (Ser 369 to Ser 410), with the exception of Ser 373, which also forms few contacts with the active site of CK1$\delta$. The N-terminus is enriched in charged amino acids (mostly positive) (Fig. 3b, c and Supplementary Fig. 6), which may hinder its binding to the CK1$\delta$ active site, since the active site features multiple charged residues and is overall positively charged (Supplementary Fig. 5a, b). However, the effect of the charges can change after the phosphorylation of Ser residues, which introduces negative charges along the sequence. On the other hand, the C-terminus has more aromatic residues[51,70] and the aromatic residues make many contacts with CK1$\delta$. (Fig. 3b, c and Supplementary Fig. 6). This difference between the N- and C-terminal segments of the TDP-43 LCD is also apparent on the correlation plot in Fig. 3d, where the N-terminal residues have both low rates and low number of contacts, whereas the C-terminal residues have mostly high phosphorylation rates and many contacts with the active site.

To ensure that our results are not overly dependent on the choice of the simulation model, we 1) repeated them with CALVADOS3 force field[48,57] and 2) investigated the effect of explicitly modeling the C-terminal helix (residues 319–341) of TDP-43 LCD (Supplementary Methods). CALVADOS3 has been parameterized to capture the interactions of disordered regions and folded proteins such as CK1$\delta$. Encouragingly, we found very similar relative values for the phosphorylation rate $r_P$ by CK1$\delta$ (Supplementary Fig. 8). Another concern is that our simulations might underestimate or overestimate interactions of the conserved region (residues 319–341) of TDP-43 LCD, which adopts helical conformations. In our initial simulations, the effect of the helix was captured only in an effective sense, through the simulation parameters of the residues in this region. Explicitly modeling the C-terminal helix by fixing residues 320–332 as rigid body (Supplementary Methods), increases contacts of Ser residues in this region with the CK1$\delta$ active site, most notably Ser 317, Ser 332, and Ser 333 (Supplementary Fig. 9), but otherwise does not affect the trends in the contact statistics of TDP-43 LCD Ser residues with the CK1$\delta$ active site. Besides the helical region Ser residues, in the C-terminal part of the LCD, Ser 369 to Ser 410 engage in the most contacts as in the simulations without explicitly modeling the helix. Phosphorylation of Ser 332 has been shown to disrupt the helical structure[71] and thus one might consider that the simulation residues without the fixed helical structure give a more accurate picture of the contact statistics of the interaction of TDP-43 with the active site of CK1$\delta$. Simulating the coupling between secondary structure and phosphorylation requires a more detailed simulation model[52,72,73].

## Prior phosphorylation alters TDP-43 phosphorylation dynamics

Although the correlation between the relative rates for CK1$\delta$ and TDP-43 contact formation and the phosphorylation rates is strong, there are deviations from this simple relationship (Fig. 3d), which could hint at possible correlations between phosphorylation events. For instance, Ser 410 forms contacts more than two times more readily than Ser 407,

but their phosphorylation rates are the same within the statistical uncertainty (Fig. 3b, c). To better understand the underlying correlations, we expanded our analysis of the phosphorylation kinetics. To estimate the phosphorylation rates $r_P$, we assumed that the phosphorylation process is a memory-less process, which follows single-exponential kinetics[74]. In this case, observing a single event is in principle sufficient to estimate the rates of a process. In addition to the number of events one observes, the time spent waiting before an event happens also contributes to the rate estimate. We checked the results by fitting the cumulative histograms of phosphorylation times for each Ser with a simple single-exponential process and an exponential process conditioned to another exponential process (e.g., the binding of TDP-43 to CK1$\delta$) (Methods). Most of the times the conditioned exponential process fits perfectly. We found that the rate extrapolations from the two fits are in agreement with the Bayesian estimates (Supplementary Fig. 10). It is interesting to notice that the fastest rate is different for every Ser (Ser 266 with a second rate coefficient of 13.5 $\mu s^{-1}$ and Ser 393 of 56 $\mu s^{-1}$) suggesting that the phosphorylation of some serines could involve other processes than the binding to CK1$\delta$, e.g., the previous phosphorylation of another Ser. For Ser 410, the two fit extrapolation and the single-exponential fit are in agreement, with differences in the phosphorylation rate of about 2%, while for Ser 403, the conditioned process fit leads to an 8% smaller rate compared to the single-exponential fit. For Ser 407, the conditioned process yields a 10% larger rate. These comparisons suggest that the phosphorylation of Ser 403 and Ser 407 could actually follow a more complex process.

We determined the most likely order of phosphorylation to understand the correlation between phosphorylation events and differences from what the contact statistics at equilibrium would predict. In order to study more deeply the phosphorylation pattern of TDP-43, we count for each pair of Ser residues (Ser$_i$, Ser$_j$) how many times Ser$_i$ is phosphorylated before Ser$_j$. We aggregate data from our 100 trajectories to compute the probability $p(t_P(\text{Ser}_i) < t_P(\text{Ser}_j))$, where $t_P(\text{Ser}_i)$ is the time of phosphorylation for Ser$_i$ from the start of the simulation. We show $p(t_P(\text{Ser}_i) < t_P(\text{Ser}_j))$ as a heat map in Fig. 3e. We see again that, on the single-molecule level, C-terminal residues are typically phosphorylated first. The lower right corner shows that on average, C-terminal residues are much more likely to be phosphorylated before N-terminal residues. As a corollary, the upper left sub-matrix shows that C-terminal residues are rarely phosphorylated after N-terminal residues. Instead, looking at the lower left block, we see that Ser 266 and Ser 273 are usually the first to be phosphorylated in the N-terminal region, while the serines within residues 333 and 350 are the last ones. In the end, by focusing on the C-terminus on the upper right block, we see that the first phosphorylations occur on Ser 369, Ser 393, Ser 395, Ser 403 and Ser 410, followed by Ser between 377 and 389 and Ser 407. In Fig. 3f, we aggregate the data from the different trajectories and illustrate the likelihood for Ser 410 of getting phosphorylated after $n$ other Ser through the thickness of the arrows pointing to pS410 from $n$p. In the figure, the state pS410 includes all the possible configurations in which Ser 410 is phosphorylated, while $n$p are the configurations with $n$ pSer different from Ser 410, as shown in the inset for four different examples of state 3p. Very often Ser 410 is among the first three residues to be phosphorylated. Only in a few trajectories, Ser 410 is phosphorylated after nine or eleven other Ser residues are already phosphorylated. Ser 395 shows similar behavior to Ser 410 (Supplementary Fig. 11). While Ser 403 and Ser 407 are also phosphorylated early on by this analysis, they are less frequently the first Ser residues to be phosphorylated compared to Ser 410 (Supplementary Fig. 11), which is in line with the deviations from single-exponential behavior (Supplementary Fig. 10). A possible influence of prior phosphorylation can also be detected for Ser 373. Ser 373 forms few contacts but is readily phosphorylated. For comparison, the phosphorylation rate of Ser 373 is just slightly lower than that of

Ser 375, which has twice as many contacts. Supplementary Fig. 11 demonstrates that Ser 373 is phosphorylated when multiple Ser residues are already phosphorylated. Notably, adjacent Ser 369 engages in many more contacts than Ser 373 and has a high phosphorylation rate. Fig. 3e shows that the probability $p(t_P(Ser369) < t_P(Ser373))$ of Ser 369 to be phosphorylated before Ser 373 is approximately 0.8. Ser 369 is also typically phosphorylated before Ser 375 is phosphorylated $(p(t_P(Ser369) < t_P(Ser375)) = 0.7)$. Changes in the interaction of CK1$\delta$ with TDP-43 LCD, as more residues are phosphorylated, could explain why phosphorylation rates are not fully accounted for by the interaction propensities of the LCD with the active site of the enzyme in equilibrium. By analyzing long equilibrium MD simulations with VAMPnet[68], we find that the phosphorylation facilitates the binding of CK1$\delta$ to the substrate TDP-43 LCD, with the binding free energy going from 5.0 kJ mol$^{-1}$ in the case of wild-type TDP-43 LCD to 1.2 kJ mol$^{-1}$ for a chain with pSer 395, pSer 403 and pSer 410 (Supplementary Note 5). As a result, the first phosphorylation events speed up further phosphorylation events, in agreement with what suggested by experiments[41], and we find that, in the simulations of enzymatic phosphorylation of TDP-43, phosphorylated TDP-43 LCD stays attached to CK1$\delta$ (Supplementary Fig. 12).

### Determinants of the pattern of TDP-43 phosphorylation by CK1$\delta$

To better understand drivers of sequence-specific phosphorylation, we altered the interactions of TDP-43 and CK1$\delta$ in our simulations. We found that the proximity of the Ser residues to the N- and C-termini does not affect the phosphorylation rates. It has been hypothesized that the tendency of C-terminal residues to get phosphorylated could be due to the greater accessibility of residues close to the N- and C-termini of a disordered protein chain[39]. In order to understand whether the phosphorylation pattern is affected by the position of the Ser residues along the TDP-43 LCD chain and not only by the neighboring residues, we repeated the same simulation but replacing all the residues of TDP-43 LCD different from Ser with an averaged interaction strength bead ("average-$\lambda$" in Fig. 3g). The averaged interaction strength bead has mass, size and hydrophobicity averaged from the wild-type TDP-43 LCD, while the electric charge is zero (Methods). Note that there are no cation-$\pi$ interactions in this case. Due to the absence of aromatic residues and cation-$\pi$ interactions, the phosphorylation rates (Fig. 3b, h), as well as the contact frequency in equilibrium (Fig. 3c, i), are one order of magnitude lower in the case of the averaged interaction sequence compared to the wild-type TDP-43. From the contact frequency $r_c$ in Fig. 3i, we can see that in equilibrium, before any phosphorylation occurs, the probability of contact is uniform along the chain, suggesting that the ends are not a priori more accessible, and hence that sequence context and its effects on molecular recognition likely explain the prominence of C-terminal TDP-43 phosphorylation. By looking at the phosphorylation rates $r_P$ (Fig. 3h), we can see that the C-terminal domain is more phosphorylated (see also Supplementary Fig. 13 left). We also computed the probability $p(t_P(Ser_i) < t_P(Ser_j))$ (Supplementary Fig. 13 right), which also demonstrated that the C-terminus is phosphorylated before the N-terminus. This suggests that the negative charges of the pSer also play a role. These are denser in the C-terminus when TDP-43 gets hyperphosphorylated. Indeed, by distributing the Ser residues at equal distances across the protein sequence ("av.-$\lambda$ equi-Ser" in Fig. 3g), phosphorylation rates are uniform within the statistical uncertainty (Fig. 3h). A similar behavior has already been found in experiments for the case of cyclin-dependent kinases phosphorylation of multisite targets[75].

In order to understand the importance of charges in the phosphorylation of TDP-43 LCD, we repeated the simulations of TDP-43 LCD, but switching off the electrostatic interactions of the residues other than pSer. In this case, the cation-$\pi$ interactions are still present. We found that the contact frequency $r_c$ without phosphorylations is

higher for the N-terminal Ser residues, while for the C-terminal, we obtained comparable results, with some Ser residues showing a small increase in the number of contacts (Fig. 3c, in red). However, the results in Fig. 3b (in red) show a decrease of phosphorylation rates $r_P$ by roughly a factor of 4 for the N-terminal phosphosites and for the C-terminal Ser residues in the proximity of aromatic residues (Ser 369, Ser 387, Ser 393, Ser 395, Ser 403 and Ser 410), while the phosphorylations of the other C-terminal Ser are even more suppressed. This means that, even though the positive charges in the N-terminus screen the interaction with the active site of CK1$\delta$ in the absence of pSer, they help the phosphorylation when some serines are already phosphorylated. By switching off the charges of residues different from pSer, we also modify the phosphorylation pattern, with the N-terminal Ser having more early phosphorylations (e.g., Ser 266, Supplementary Fig. 14 left) and some of the C-terminal ones being disfavored (e.g., Ser 389, Supplementary Fig. 14 right), as shown in Supplementary Fig. 15 left. Notice that in the case of TDP-43 without charges, $r_c$ and $r_P$ are highly correlated (with correlation coefficient of 0.9, Supplementary Fig. 15 right), which highlights how the charges of TDP-43 contributes to the complexity of the phosphorylation pattern.

### CK1$\delta$ phosphorylates TDP-43 in condensates

Next, we wondered how condensation of TDP-43 LCD would change the interactions of CK1$\delta$ and TDP-43 and found that CK1$\delta$ binds to the condensates, eventually hyperphosphorylates TDP-43 which dissolves the TDP-43 LCD condensates (Fig. 4a). Recent experiments have shown that hyperphosphorylation of TDP-43 can prevent phase separation and aggregation by increasing the solubility of TDP-43[39,71]. However, it remains unclear whether kinases, such as CK1$\delta$, actually bind to TDP-43 condensates or only phosphorylate TDP-43 in dilute solution. Snapshots from an example simulation with five CK1$\delta$ enzymes are shown in Fig. 4a (Supplementary Movie 3), with the first snapshot depicting the starting configuration with 200 TDP-43 chains in phase-separated condensate and the enzymes (blue molecules) randomly placed in the box. After 1 μs of simulation time ($1 \times 10^8$ steps), when about 24% of the Ser residues have already been phosphorylated, the enzymes are all attached at the surface of the condensate, and they are in the process of phosphorylating more serines (pSer in red). Hyperphosphorylated TDP-43 chains start to disassociate from the condensate, which appears almost entirely dissolved after 5 μs of simulation time ($5 \times 10^8$ steps) in the last snapshot. We report in Fig. 4b the percentage of chains in the condensate (blue, left y-axis) and the percentage of phosphorylated Ser (red, right y-axis) over time from simulations with 1, 3, and 5 enzymes, averaged over 4 independent replicas. The percentage of TDP-43 chains in the condensate drops over time as the phosphorylation count increases. We computed the size of the condensate using a standard clustering analysis algorithm, which enabled us to track the chains in the largest cluster at every frame (Methods). In every simulation, the condensate starts to lose TDP-43 chains when about 24-25% of Ser are phosphorylated. As a result, the speed of phosphorylation decreases with time, as the TDP-43 chains start to migrate in the dilute regime, far from the action of the enzymes. This effect is particularly evident in the case of 5 CK1$\delta$ after about 3.5 μs. It is interesting to notice that the speed of phosphorylation decreases slightly with time, even before the beginning of the dissolution, with C-terminal Ser being the most affected (Supplementary Fig. 16, upper panel). The slowing down of the phosphorylation rates of the most accessible Ser suggests a possible saturation effect. We also notice that, at least after about 4.5% of Ser are phosphorylated, most of the TDP-43 chains feature less than 4 phosphates, with a small minority of chains being hyperphosphorylated (Supplementary Fig. 17, blue). Moreover, the action of the kinases is limited to the surface of the droplet, and the density of pSer residues in the condensate remains higher at the interface, supporting the idea of an early saturation of

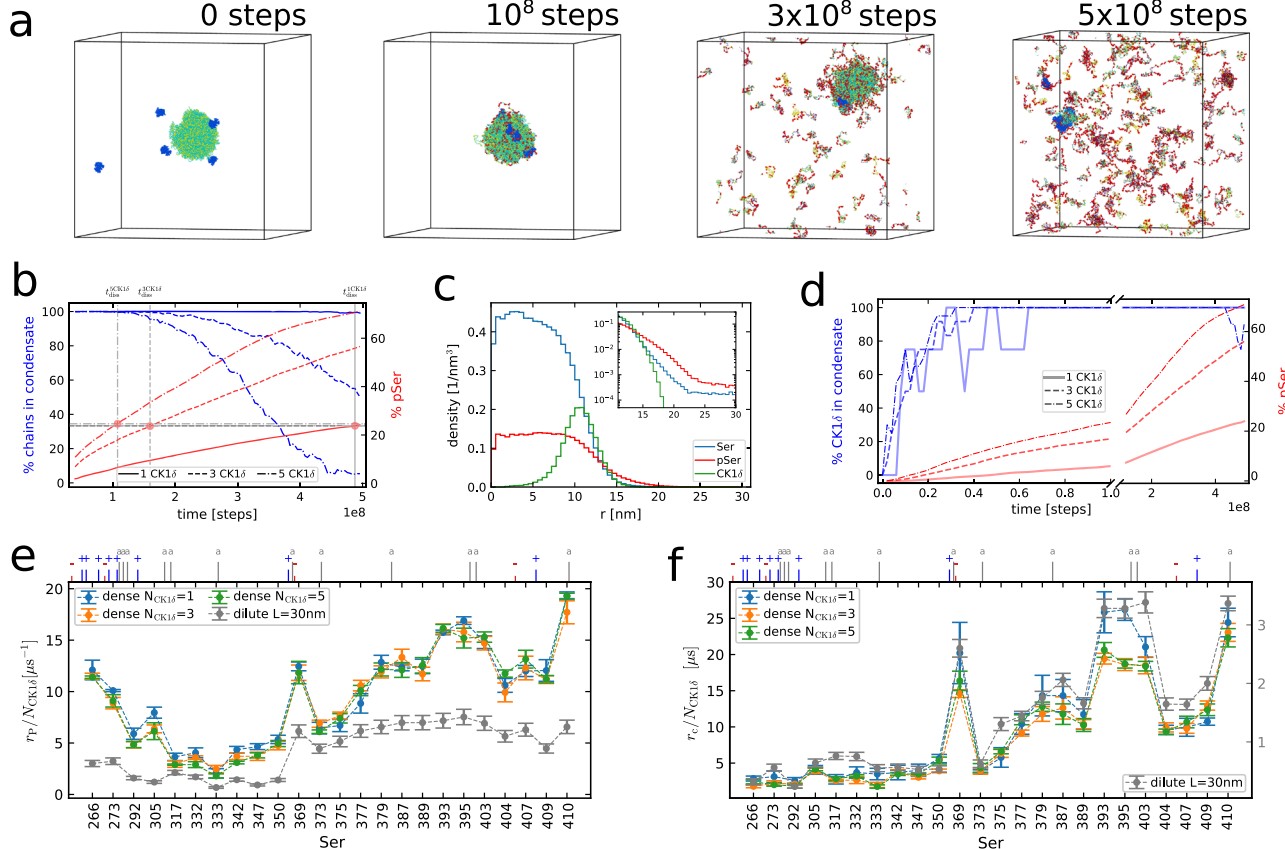

**Fig. 4 | Analyzing the effect of hyperphosphorylation of a TDP-43 LCD condensate and the interaction of CK1δ with the condensate.** All the simulations involved in the plots are performed in a cubic box with a 100 nm side length with 200 TDP-43 LCD chains. **a** Snapshots from simulation with 5 CK1δ at times 0, 1, 3, and 5 μs showing the dissolution of the TDP-43 condensate. The enzymes are colored blue, and the phospho-serines red. **b** Percentage of TDP-43 chains in the condensate (blue, left *y*-axis) and percentage of phosphorylated Ser (red, *y*-axis) over time for simulations with 1, 3, and 5 CK1δ. **c** Radial density profile of Ser (blue), pSer (red), and CK1δ (green) residues from equilibrium simulation with 40% of phosphorylated serines and 5 CK1δ. Counts are normalized by the volume of the spherical bin. In the inset, zoom on the condensate interface area. **d** Percentage of

CK1δ attached to the condensate (blue, left *y*-axis) and percentage of phosphorylated Ser (red, right *y*-axis) over time for simulations with 1, 3, and 5 CK1δ. **e** Mean values +/− SEM of phosphorylation rates $r_P$ for every Ser of TDP-43 LCD divided by the number of CK1δ chains from 4 independent condensate simulations in the presence of 1,3, and 5 enzymes. Results for the single-chain case reported in gray (from Fig. 3b). The ticks on top show the position of the charged (blue `+' positive, red `−' negative) and aromatic (gray `a') residues. **f** Mean values +/− SEM of contact rates $r_c$ for every Ser of TDP-43 LCD divided by the number of CK1δ chains from 4 independent condensate simulations in the presence of 1, 3, and 5 enzymes at equilibrium. Results for the single-chain case reported in gray (from Fig. 3c). Raw data provided in the Source Data file.

the most accessible phosphosites. We computed the radial density profile of Ser, pSer and CK1δ residues from an equilibrium simulation with 40% of phosphorylated serines and 5 kinases by normalizing the counts of residues per spherical bin with the volume of the bin (Fig. 4c). Also in this case, the condensate is detected through a clustering algorithm. Most CK1δ residues are located at the interface of the condensate (at around 11 nm from the center). By looking at the inset in semi-log scale, we notice that the density of pSer is higher than the Ser one at distances greater than 14 nm, reaching 25% of the total number of beads in the bins for r ≥ 18 nm (Supplementary Fig. 18 left). Indeed, the hyperphosphorylated chains often stretch out of the condensate before leaving it, as visible in Fig. 4a at $3 \times 10^8$ MD steps. The reason why CK1δ chains locate on the surface of the condensate might be linked to the higher hydrophobicity of TDP-43 LCD. Indeed, if we switch off the rescaling of the hydrophobic interactions (Methods) for CK1δ, we observe that the enzyme locates in the interior of the condensate (Supplementary Fig. 19). It is also interesting to observe that in our simulations TDP-43 chains mostly leave the condensate when an average of 16 Ser residues (out of 24) are phosphorylated within the same chain, as shown in Supplementary Fig. 21, where we report the distribution of the number of pSer per chain in dilute phase at different times through a violin plot.

TDP-43 phosphorylation facilitates CK1δ binding to the condensate compared to unphosphorylated condensates at equilibrium. In Fig. 4d, we show the percentage of enzymes attached to the condensate over time compared to the percentage of phosphorylated Ser, averaged again over 4 replicas. With increasing phosphorylations, CK1δ binds more stably to the condensate, with every enzyme in all our simulations being attached to the condensate between 15% and 60% pSer. Above 60% pSer, some kinases start to leave the droplet, which is almost completely dissolved (as in Fig. 4a at $5 \times 10^8$ MD steps). Instead, in equilibrium simulations without phosphorylation only about 35% of enzymes are attached to the droplet on average (Supplementary Fig. 18, right). These results suggest that the negative charges of the pSer residues enhance the binding to the enzyme positively charged residues, similarly to the single-chain case. Even in equilibrium simulations at 40% pSer with 1 CK1δ chain placed outside the condensate, we observe that the enzyme binds to the condensate rather than remaining bound to the hyperphosphorylated chains in dilute phase (Supplementary Fig. 20, Movie 6). This behavior might be due to the higher density of pSer on the surface of the condensate than in the dilute phase (Fig. 4c). We computed the energy contribution per CK1δ residue for electrostatic interactions (Yukawa pair potential), hydrophobic interactions (Ashbaugh-Hatch pair potential), and cation-π

interactions separately from equilibrium simulations both without pSer (Supplementary Fig. 22, upper panel) and with 40% pSer (Supplementary Fig. 22, lower panel) from 30 uncorrelated frames. The hydrophobic interactions are in general, the dominant ones across the enzyme chain, with average strength per residue of $(-0.117 \pm 0.006)$ kJ mol$^{-1}$ in the case without pSer and $(-0.209 \pm 0.008)$ kJ mol$^{-1}$ in the case with 40% pSer. The cation-$\pi$ interactions are also important, even though more localized, reaching strengths lower than $-0.4$ kJ mol$^{-1}$ for some residues. However, the most relevant difference is in the electrostatic interactions, with positively charged residues going from a mean positive value of the potential energy in the absence of pSer, which oppose the cation-$\pi$ interactions, to strong negative values for the case with 40% pSer (the opposite happens to the negatively charged residues). This reflects in the average strength per residue, going from $(+0.006 \pm 0.003)$ kJ mol$^{-1}$ without pSer to $(-0.17 \pm 0.06)$ kJ mol$^{-1}$ with 40% pSer.

In our simulations, the protein sequence context determines how much a given Ser residue is phosphorylated in the condensates. We compute the phosphorylation rates $r_P$ for each Ser of TDP-43 LCD from the counts of phosphorylations, and we compare them with the single-chain simulation results. For this computation, we use only the part of the simulations before the start of the condensate dissolution. We can see from Fig. 4e that the phosphorylation rates scale proportionally with the number of enzymes in the box. Moreover, the phosphorylation of the N- and C- terminal serines (namely Ser 266 and Ser 410) is enhanced, as well as for Ser 393, Ser 395, and Ser 403, compared to the single-chain case (see correlation plot in Supplementary Fig. 23 left).

The probability of contact with the active site of CK1$\delta$ in the condensate for every Ser of TDP-43 LCD differs from the single-chain case roughly by a factor 6. The C-terminus is more accessible, in particular Ser 369, Ser 393, Ser 395, Ser 403, and Ser 410, as shown in Fig. 4f, similar to what occurs in the dilute regime. For the dense phase, the phosphorylation rates seem very well correlated to contact rates in equilibrium for the C-terminal serines (sample Pearson correlation 0.91), while the end of the N-terminus is more phosphorylated compared to what one would expect based on the contact statistics from equilibrium simulations (Supplementary Fig. 23, right).

Overall, our simulations show that CK1$\delta$ binds to TDP-43 condensates, and as TDP-43 is phosphorylated by CK1$\delta$, condensates progressively dissolve. In analogy to simulations of phosphorylation of TDP-43 by CK1$\delta$ in dilute solution, contact statistics largely account for the phosphorylation rates. In our simulations, the interactions of the helical region of TDP-43[18–20] are determined by the interaction parameters of the residues of this region, which is enriched in hydrophobic residues[15,24]. As for simulations in the dilute phase (Supplementary Fig. 9), we also checked that we obtained similar results by modeling the helical structure in residues 319–341 explicitly (Supplementary Fig. 24 and Supplementary Methods). Dissolution of the TDP-43 condensates in simulations with 5 CK1$\delta$ chains, with more and more chains outside the condensate, starts when about 24% of Ser are phosphorylated (Supplementary Fig. 24), which is similar to simulations without enforcing the helical structure.

## Role of CK1$\delta$ IDR in TDP-43 phosphorylation

In simulations with full-length CK1$\delta$, including the disordered region (IDR) from residue 295–415, we find that the phosphorylation behavior is similar to the case without IDR both for the single TDP-43 single CK1$\delta$ chains simulations (Fig. 5a, Supplementary Movie 4) and in condensates (Fig. 5b and Supplementary Movie 5). The tail helps the recruitment of the substrate TDP-43 LCD (see insets in Fig. 5a, b), but it may induce a modification in the structure of the folded domain, as predicted by AlphaFold2[76] (Supplementary Fig. 25), which reduces the accessibility of the active site residues compared to the folded domain structure from X-ray crystallography[77]. We can see in Fig. 5c that the

full-length CK1$\delta$ (red) has in general, more contacts in equilibrium simulations without pSer than the CK1$\delta$ without tail (yellow) in the residues of the folded domain (from 0 to 294), due to the stronger binding of TDP-43 LCD with the CK1$\delta$ IDR. Indeed, the binding free energy goes from 5 kJ mol$^{-1}$, in the case of single CK1$\delta$ folded domain and single non-phosphorylated TDP-43, to $-4$ kJ mol$^{-1}$ for the case of closed full-length CK1$\delta$, as analyzed by VAMPnet[68]. However, the active site features a comparable amount of contacts in the two cases, with residue 149 having even more contacts in the simulations without tail, due to the lower accessibility to the active site for the full-length CK1$\delta$ with closed configuration. By looking instead at the disordered domain of CK1$\delta$ (residues from 295 to 415), we find that the tail has more contacts with TDP-43 LCD compared to the folded domain surface residues. Similar results are obtained from the equilibrium simulations with the TDP-43 condensate (Fig. 5d). In this case, the overall higher contact frequency is due to the higher density of TDP-43 chains in the condensate compared to the dilute case. In this case, the strong interactions between CK1$\delta$ IDR and TDP-43 chains allow the enzyme to be anchored to the surface of the condensate even in the absence of pSer residues. In Fig. 5e, we can see that the folded domain residues are localized on the surface of the condensate, at around 11 nm from its center, in equilibrium simulations with 40% pSer (as in the case without tail), while the IDR residues enter the interior of the condensate. As for the case without tail, the density of pSer residues is higher than the density of Ser residues at the interface (inset of Fig. 5e).

To assess the effect of the conformational change of the folded domain, we simulated full-length CK1$\delta$ both with the folded domain structure from AlphaFold2 (referred to as closed configuration) and with the one from experiments (open configuration)[77]. We compared the results for the phosphorylation rates $r_P$ and the rate of active-site contact formation $r_c$ between the simulations with full-length CK1$\delta$ with open and closed configurations and with CK1$\delta$ folded domain. We can see in Fig. 5f, g that the C-terminal Ser residues still have higher $r_P$ and $r_c$ in single-chain simulations compared to the N-terminus. The full-length CK1$\delta$ with open configuration has much higher $r_c$ compared to the CK1$\delta$ folded domain, because of the recruitment effect of the tail. This also translates into a higher phosphorylation rate of the C-terminal serines. Simulations with closed full-length CK1$\delta$, instead, result in a reduction of $r_P$ and $r_c$ by one order of magnitude compared to the case without a tail. The closed configuration of the folded domain also reduces the phosphorylations of Ser 266, Ser 273, and some of the C-terminal Ser residues relatively to the other serines, leaving Ser 369, Ser 387, Ser 393, Ser 395, Ser 403, and Ser 410 being the most phosphorylated. The values of $r_P$ and $r_c$ are more correlated in this case than for the folded domain alone (Supplementary Fig. 26 left). Similarly to the case without tail, TDP-43 stays bound to the full-length CK1$\delta$ after 2-3 phosphorylations (Supplementary Fig. 27).

In simulations with TDP-43 condensate, $r_P$ and $r_c$ increase compared to the dilute case, because of the high amount of chains, and they are proportional to the number of enzymes in the simulation, for both the cases with closed (Fig. 5h, i) and open configurations (Fig. 5j, k). Also in condensate, $r_P$ and $r_c$ are one order of magnitude smaller in the case of closed full-length CK1$\delta$ compared to the CK1$\delta$ folded domain. Contacts and phosphorylation counts are also highly correlated in this case, since the abundance of chains allows the enzyme to neglect the less accessible phosphosites and phosphorylate the most accessible ones from each chain (Supplementary Fig. 26 right). This disfavors the phosphorylation of Ser 389, Ser 404, Ser 407, and Ser 409, which are less phosphorylated compared to the dilute case. Instead, for full-length CK1$\delta$ with open configuration, all the phosphorylation rates are comparable to those from the simulations without tail (Fig. 5j), despite the rates of contact in absence of phosphorylations are roughly 3-fold higher (Fig. 5k). The tail allows the enzyme to stably bind to the condensate even in absence of phosphorylation, unlike the case with CK1$\delta$ folded domain, increasing the

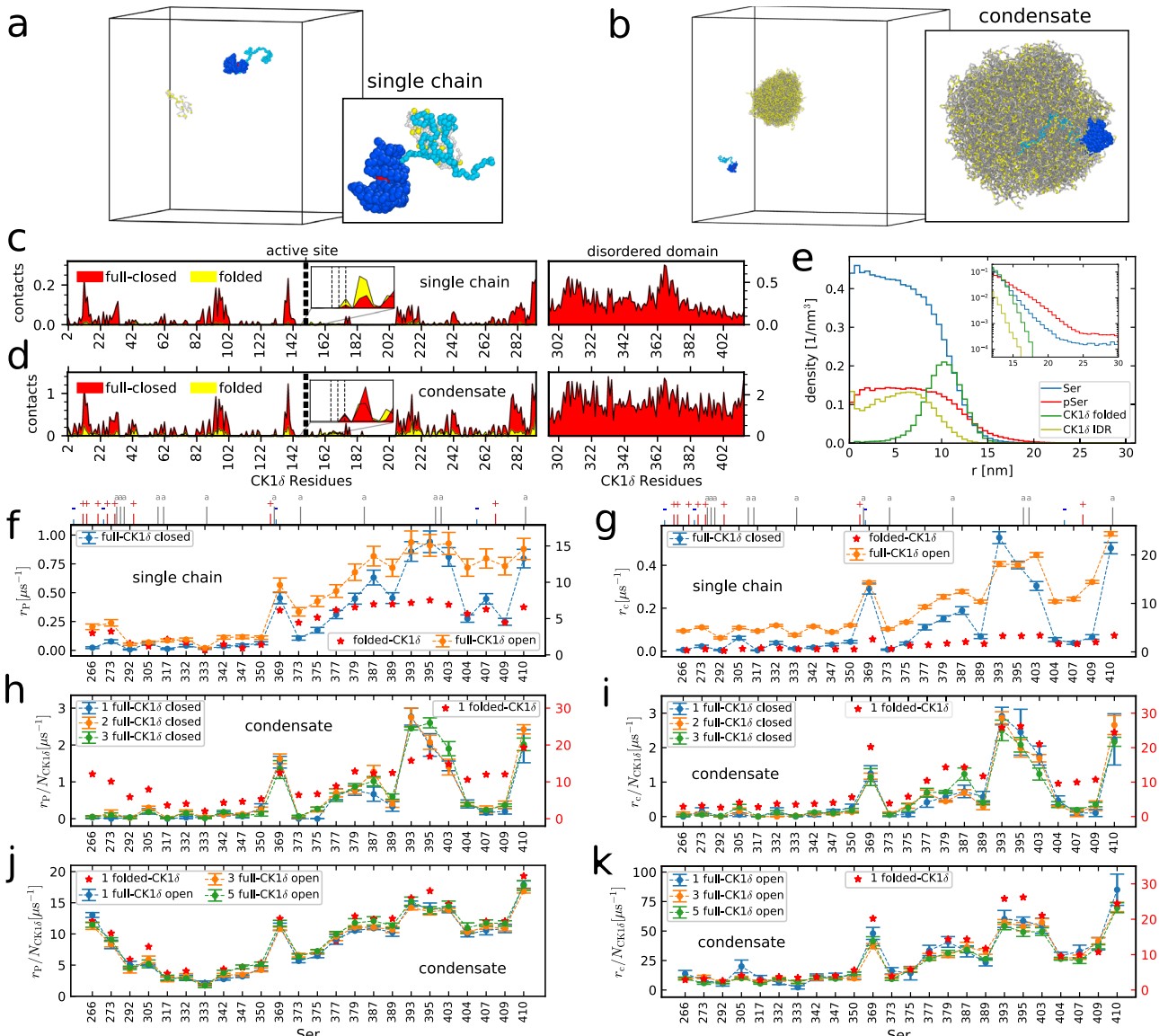

**Fig. 5 | Role of CK1δ IDR in TDP-43 phosphorylation. a** Example of simulation setup with full-length CK1δ (blue) and TDP-43 LCD (gray, Ser in yellow) in single-chain simulations. Inset: TDP-43 interacting with CK1δ IDR (light blue, active site in red). **b** Example of simulation setup with full-length CK1δ and condensate of 200 TDP-43 chains. Inset: CK1δ IDR (light blue) anchoring the folded domain (dark blue) to the surface of the condensate. **c, d** Contacts per frame for residues of full-length CK1δ closed configuration (red) and CK1δ without tail (yellow) in single TDP-43 chain simulations (**c**) and condensate (**d**) at equilibrium. Insets: contacts in the active site region (residues 149–151). **e** Radial density profile of Ser (blue), pSer (red), and full-length CK1δ (folded domain in darker green, IDR in olive green) residues from equilibrium simulation with 40% phosphorylated serines and 5 CK1δ. Inset: zoom on the condensate interface region. **f, g** Mean values +/− SEM of $r_P$ (**f**)

from 100 replicas and $r_c$ (**g**) from 30 replicas for every Ser of TDP-43 LCD in the presence of full-length CK1δ from single-chain simulations. Results with open folded domain configuration (orange) and without tail (red stars, from Fig. 3b, c) are reported on the right y-axis for comparison. The ticks on top show the position of the charged (blue `+' positive, red `−' negative) and aromatic (gray `a') residues. **h, i** Mean values +/− SEM of $r_P$ (**h**) from 4 replicas and $r_c$ (**i**) from 4 replicas with full-length closed CK1δ in condensate. Results from simulations without tail are reported in red on the right y-axis (from Fig. 4e, f). **j, k** Mean values +/− SEM of $r_P$ (**j**) from 5 replicas and $r_c$ (**k**) from 5 replicas with full-length open CK1δ in condensate. Results from simulations without tail are reported in red (right y-axis in **k**). Raw data provided in the Source Data file.

chances of contacts between Ser residues and active site. On the other hand, the anchoring effect of the tail is not evident in the phosphorylation rates, since even the enzymes without tail stay attached to the condensate surface after some initial phosphorylations events.

In experiments, truncated CK1δ is more active than the full-length enzyme[43]. In the AlphaFold2 prediction, the CK1δ IDR binds to the surface of the folded domain in proximity to the active site, probably inducing the closed configuration and occluding the active site (Supplementary Fig. 25). However, in our coarse-grained simulations we find that the IDR rarely forms close contacts with the active site and any close contacts are lost very rapidly, both in single-chain simulations

and in condensate (Supplementary Fig. 28,29). Nonetheless, the conformational change induced by the CK1δ IDR on the active site reduces the phosphorylation rates even in coarse-grained simulations, compared to simulations where the active site was in a more open conformation, in line with experiments (Fig. 5f).

## Discussion

We have demonstrated how to generate Markov-state models from molecular simulations of chemically-driven non-equilibrium steady states (NESS) and thus verify the thermodynamic consistency of simulations of chemically-driven dynamics. Chemically driven NESS

are essential in cell biology[1]. Cells require the constant turnover of fuels[8,9] and metabolites to grow and thrive. Chemically-driven NESS are likely also essential in the function of biomolecular condensates in cells[2]. Molecular simulations and Markov state models will be important to resolve how chemically-driven dynamics underpins biological processes[4]. In our simulations, ATP and ADP are modeled implicitly, but the interactions of ATP can modulate interactions of proteins in condensates[8,9], and one can envisage to explicitly simulate ATP and ADP to capture such effects in coarse-grained molecular dynamics simulations. We anticipate that our approach to establish the thermodynamic consistency of simulations and the combination of molecular dynamics and Monte Carlo can be readily applied to more complex systems and simulations of such systems in high resolution[54,63]. For more complex systems, extracting kinetically meaningful states becomes even more challenging. In this respect, advances based on neural networks and Koopman theory are highly encouraging[68,69,78,79].

PTMs such as phosphorylation of proteins are a fundamental regulatory mechanism in cells, and with molecular simulations, we can start to investigate how protein sequence and structure determine substrate-enzyme interactions and PTM patterns. Our simulations demonstrate that the IDR of CK1δ could have important roles in TDP-43 phosphorylation, (1) by facilitating the binding to condensates and (2) by inducing a closed configuration of the folded domain, as predicted by AlphaFold2, which inhibits the enzyme in our simulations, in line with experiments[43,44]. It is important to note that details of the conformations of proteins will be critical for the molecular recognition of potential phosphorylation sites by kinases, and more detailed and all-atom molecular simulations[20,73] and high-resolution experiments[20] will be required to fully understand the recognition mechanisms. More detailed simulation models[20,51,52,73] could also capture how secondary structure changes upon phosphorylation, as experimentally shown for the C-terminal helix of TDP-43[71]. However, the comparison with CALVADOS3 force field (Supplementary Fig. 8), which employs elastic network rather than rigid body dynamics for the folded domains, is encouraging and suggests that coarse-grained models capture -at least in part- sequence-specific interactions.

Overall, our simulations point to a potential preference for the C-terminal residues of TDP-43 on account of its sequence. Aggregated TDP-43 in patient samples is frequently phosphorylated at, e.g., Ser 379, Ser 403/Ser 404, and Ser 409/Ser 410[35,38], which are among the most phosphorylated residues also in our simulations. Importantly, our simulations show how phosphorylation events are interdependent. Phosphorylation of one Ser residue can predispose the enzyme to phosphorylate an adjacent Ser residue[41]. For instance, the phosphorylation of Ser 407 seems to be driven by the earlier phosphorylation of Ser 410. We also highlight how both serine distribution and presence of charges enhance the phosphorylation of C-terminal Ser residues (Fig. 3b, h). However, aromatic residues, and in general interaction-prone residues, in the C-terminus[24] favor the formation of contacts with the active site of the kinase even in simulations of TDP-43 LCD without charges and phosphorylated serines (Fig. 3b), which shows once more the importance of the sequence context.

Due to the high concentration of substrates in condensates, proteins are readily phosphorylated. The phosphorylation rates for Ser residues are higher for TDP-43 in condensates than in the dilute phase. Interestingly, phosphorylation patterns are overall similar in dilute solution and condensates. While there are differences in the phosphorylation propensities, sequence context still determines which sites can be phosphorylated. One can speculate that differences between phosphorylation in dilute and dense solutions could be partly explained by the overall higher phosphorylation level in condensates, which means that some sites will effectively be more readily phosphorylated than in dilute solutions[42], while the kinase may retain sequence-dependent recognition of substrates in the condensates.

Our simulations suggest that enzymatic phosphorylation of TDP-43 by CK1δ can occur in condensates, which is a step towards elucidating the molecular consequences of this PTM and, in particular, phosphorylation patterns detected in patient samples. Experiments will be required to test whether the enzymatic phosphorylation of TDP-43 in condensates dissolves TDP-43 condensates and may protect against the formation of insoluble TDP-43 aggregates. This may be a key mechanism in the early stages of the formation of aggregates in neurons[17]. TDP-43 hyperphosphorylation was previously shown to reduce TDP-43 condensation and aggregation[39], but whether enzymes phosphorylated TDP-43 in dilute solution or in the condensates was not known. Our simulations show that partially phosphorylated TDP-43 condensates are stable. One can speculate that TDP-43 phosphorylation might change how TDP-43 and TDP-43 condensates in the nucleus interact with their cognate binding partners[15,17,40] and how aberrant TDP-43 condensates in the cytoplasm interact with cytoplasmic proteins[24], which could lead to toxic gain-of-function or loss-of-function[16].

TDP-43 phosphorylation could also reduce its nuclear localization and induce its accumulation in the cytoplasm, which is associated with neurodegenerative disease[20,24,39]. Recently, it was demonstrated that mutations in the C-terminal region of TDP-43, which reduce the stability of the C-terminal helix (residues 319–341), also reduce nuclear localization of TDP-43[20]. Hence, the C-terminal phosphorylations (e.g., Ser 332) we investigated could trigger the cytoplasmic localization of TDP-43 by disrupting the structure of the C-terminal helix, as has been shown in biophysical studies of the TDP-43 LCD[18,19,71].

## Methods

### Coarse-grained MD simulations

In our work, we simulated TDP-43 LCD and the kinase CK1δ using a one-bead-per-residue coarse-grained model called the hydrophobicity scale model[47] (HPS model) and a modified version of it, referred to as the modified HPS model in the text (Supplementary Methods). In these models, the water solvent and the ion concentration are implicit in the pair potential definition, which assumes an ionic strength of approximately 100 mM. We used the original HPS for the thermodynamic consistency validation simulations. For the other simulations, we used the modified HPS model in which cation-π interactions are enhanced[80], which was previously shown by Tejedor et al. to capture the relative propensity of full-length and LCD TDP-43 to phase-separate[46]. In the simulations, folded domain interactions are reduced by 30%[47,81,82] (Supplementary Methods). Simulations were conducted using Langevin dynamics at a temperature of 300 K, with integration step of 0.01 ps, friction coefficient of 0.001 ps$^{-1}$ and in a cubic box with periodic boundary conditions of side length 50 nm for the thermodynamic consistency validation simulations, 30 nm for the rest of the single TDP-43 chain simulations and 100 nm for the condensate simulations. The initial position of the chains in each replica of the single-chain simulations is randomly chosen. The equilibration time in these simulations is negligible for the force field used. Instead, for condensate simulations, each initial configuration is generated by first equilibrating the TDP-43 chains and subsequently placing the enzyme chains randomly in the dilute phase. Moreover, no additional equilibration is performed in simulations with phosphorylation reaction, in order to mimic in vitro kinase assays. The simulated TDP-43 LCD includes residues from 261 to 414 of the full-length TDP-43. The folded domain of CK1δ (residues from 3 to 294) follows a rigid body dynamics with a rotational drag coefficient of 4 ps$^{-1}$ for every axis. For simulations with CK1δ truncated at residue 294, the structure of the folded domain is provided by https://www.rcsb.org/structure/6ru7 (X-ray crystallography)[77]. For simulations with full-length CK1δ, the folded domain structure is fixed according to https://alphafold.ebi.ac.uk/entry/P48730 (AlphaFold2 prediction). Since AlphaFold2

predicts a more closed configuration of the active site for the case with IDR, we also simulated the full-length CK1$\delta$ with an open active site configuration by attaching the disordered tail to the folded domain structure from X-ray crystallography. In our simulations, CK1$\delta$ is phosphorylated, and we do not allow possible autophosphorylations of the CK1$\delta$ IDR[44].

For the single TDP-43 single CK1$\delta$ simulations (Figs. 3, 5), 100 replicas with phosphorylation reaction step and without reservoir exchange step were run, each $2 \times 10^8$ MD steps long (2 $\mu$s in simulation time) for the cases with 1 wild-type TDP-43 LCD and 1 CK1$\delta$ folded-domain, with 1 TDP-43 LCD without charges and 1 CK1$\delta$ folded-domain and 1 wild-type TDP-43 LCD and 1 full-length open CK1$\delta$, and $4 \times 10^8$ MD steps long (4 $\mu$s in simulation time) both for the case with 1 averaged-interaction polymer and 1 CK1$\delta$ folded-domain and for the case with 1 wild-type TDP-43 LCD and 1 full-length closed CK1$\delta$. To characterize the intrinsic affinity of the enzyme for TDP-43 LCD, we repeated the same simulations, but without phosphorylation reactions at thermodynamic equilibrium. We collected 30 replicas of 15 $\mu$s each for the case of wild-type TDP-43 LCD with CK1$\delta$ folded-domain, 30 replicas of 10 $\mu$s each for TDP-43 LCD without charges with CK1$\delta$ folded-domain, and for the full-length open CK1$\delta$, and 30 replicas of 30 $\mu$s each for the averaged-interaction polymer and for the full-length closed CK1$\delta$. The averaged interaction polymer is built by substituting the TDP-43 LCD residues different from Ser with a bead having zero electric charge and average TDP-43 LCD mass (98.957 amu), size parameter $\sigma$ (0.54331 nm) and hydropathy parameter $\lambda$ (0.64039) (1) leaving the serines at their original positions ("average-$\lambda$" in Fig. 3g) and (2) spreading them equally spaced along the chain ("av.-$\lambda$ equi-Ser" in Fig. 3g). Instead, in simulations with TDP-43 without charges, we switched off the electrostatic interactions of the residues different from pSer in TDP-43.

We also simulated a condensate of 200 TDP-43 LCD chains (Figs. 4, 5). We ran 4 replicas of $5 \times 10^8$ MD steps each (5 $\mu$s in simulation time) with phosphorylation steps without reservoir exchange step (as for the single-chain simulations) for the cases with 1,3 or 5 CK1$\delta$ folded-domain chains and with 1,2 or 3 full-length CK1$\delta$ with closed active site, and 5 replicas for the cases with of 1,3 or 5 full-length CK1$\delta$ with open active site. We repeated the same simulations, but without phosphorylation reactions at thermodynamic equilibrium, collecting 4 independent 5 $\mu$s long replicas for the cases with CK1$\delta$ folded-domain chains and full-length CK1$\delta$ with closed active site, and 5 independent 1 $\mu$s long replicas for the case with full-length CK1$\delta$ with open active site. For the estimate of the contact rates $r_c$, we excluded the first 200 ns from each replica, in order to account for the equilibration of the binding/unbinding process between enzyme chains and condensate. The simulations with 40% phosphorylated serines were run at equilibrium starting from a snapshot with 200 TDP-43 chains in a dissolving condensate (with 1920 pSer out of 4800) and 5 CK1$\delta$ folded-domain chains, for a total of 5 replicas of 1 $\mu$s.

All the simulation setups are reported in Supplementary Table 3. Initial and final coordinate files are available on Zenodo[83], together with some example trajectories. All the simulations involved in this paper were performed using the Python packages HOOMD-blue version 3.8.1[84] and GSD version 2.8.1. The code used for the simulations and the HOOMD-blue plugin with the Ashbaugh-Hatch pair potential for the non-bonded interactions are available on GitHub (https://github.com/ezippo/hoomd3_phosphorylation, https://github.com/ezippo/ashbaugh_plugin) and Zenodo[83,85,86].

## Phosphorylation reaction through a Monte Carlo step

In addition to the standard MD simulation, we added a Monte Carlo step to mimic the phosphorylation reaction. Every 200 steps of the MD simulation, we check if one of the TDP-43 phosphosites is in contact with the active site of CK1$\delta$, the area of the enzyme that catalyzes the reaction, identified with the residues Asp149, Phe150,

and Gly151. The contact criterion is the following: the three distances between the TDP-43 phosphosite and the residues of the CK1$\delta$ active site must all be less than 1 nm; in case more than one phosphosite is in contact with the active site at the same time step, only the closest one is taken into account. When a contact occurs, we try to switch the Ser in contact to pSer (or the opposite) with a Metropolis-like acceptance probability in Equation (2). The reverse reaction, that is the exchange of pSer with Ser, can also occur with probability $A(\text{pSer}, \text{Ser}) = \min(1, \exp(\beta \Delta U_P + \beta \Delta \mu_P))$, but it is less likely to happen when there is a chemical potential difference favouring the protein phosphorylation. ATP, ADP are modeled implicitly and are not explicitly simulated, with concentrations kept fixed and fully characterized through the choice of $\Delta \mu_P$.

The chemical potential difference in a reaction in units of $RT$ is given by the logarithm of the product to substrate concentration ratio. Considering that the ATP concentration in cells is around 1 mM, the concentration of ADP is around 10 $\mu$M, and fixing a temperature $T = 300$ K for our simulations, we get a chemical potential difference for a phosphorylation reaction $\Delta \mu_P = \mu_{ADP} - \mu_{ATP} \simeq -11.5$ kcal/mol $\simeq -48$ kJ mol$^{-1}$ (Supplementary Note 4). Observe that the ATP concentration is two orders of magnitude larger than the ADP concentration, leading to a large negative $\Delta \mu_P$ that favors the exchange of Ser to pSer and disfavors the opposite reaction. Moreover, we can mimic the ATP to ADP concentration ratio by changing the chemical potential difference in our simulation at fixed temperature. We used $\Delta \mu_P = 0, -5, -10$ kJ mol$^{-1}$ for the validation of the thermodynamic consistency simulations and $\Delta \mu_P = -48$ kJ mol$^{-1}$ for all the other simulations in the dilute regime and condensate.

## Dephosphorylation step

In our validation simulations, we assume the exchange between TDP-43 and phosphorylated TDP-43 happens without chemical driving and with equilibrium concentrations, through another Metropolis-like step (reservoir exchange step). Every 200 MD steps, we check if the distances between the TDP-43 phosphosite and the 3 residues of the CK1$\delta$ active site are larger than 25 nm (half box side length). In that case, we randomly swap the pSer of the phosphorylated TDP-43 with a Ser (or the opposite) with a Metropolis-like acceptance probability:

$$A_D(\text{pSer}, \text{Ser}) = \min(1, \exp(-\beta \Delta U_D)) \quad (4)$$

where $\Delta U_D$ is again the difference between the potential energy of the configuration with Ser and the one of the configuration with pSer. In this case, there is no chemical driving force, the reaction obeys detailed balance, and thus it does not inject any additional energy into the system. This exchange step mimics a larger reservoir of TDP-43 and phosphorylated TDP-43, and thus enables us to simulate multiple phosphorylation cycles at the level of a single enzyme and single substrate protein simulation.

## Thermodynamic consistency validation simulations

We simulated the system with one CK1$\delta$ and one TDP-43 LCD with only one reactive residue. We repeated the simulation for 6 different reactive serines along the TDP-43 LCD, i.e., Ser292, Ser 317, Ser 369, Ser 387, Ser 403, Ser 409, and $\Delta \mu_P = 0, -5, -10$ kJ mol$^{-1}$. Simulations were conducted for 20 $\mu$s in a cubic box of 50 nm side length with periodic boundaries using HPS model force field. In order to get better statistics, we took 4 replicas of 25 $\mu$s long trajectories for Ser 403 and $\Delta \mu P = 0, -5$ kJ mol$^{-1}$. For the estimate of the MSM transition rates, we excluded the first 100 ns of equilibration from each replica (of the same order of the lag time) to ensure the independence of the first MSM step from the initial configuration. We used these longer trajectories for the estimates of $\Delta \mu_{cycle}$ with different lag times and with the version of VAMPnet with more input distances. Error bars on $\Delta \mu_{cycle}$

were obtained via bootstrapping of the total simulation trajectory collected.

## VAMPnet architecture and training

For the bound state recognition, we performed a nonlinear dimension reduction using a neural network with two identical lobes, following the VAMPnet architecture and the hyper-parameter optimization used by Mardt et al.[68]. Each lobe is composed of an input layer with 154 nodes, one for each residue of TDP-43 LCD, one hidden layer with 30 nodes that employs exponential linear units (ELU) and an output layer with 2 nodes, ideally bound and unbound state, and a final Softmax classifier to obtain probabilities of bound and unbound configurations as output. As input for the neural network, we used the 154 distances between each residue of TDP-43 LCD and the active site of CK1$\delta$. We chose a learning rate of $0.5 \times 10^{-2}$ and a batch size of $4 \times 10^4$. The neural network was trained for 100 epochs on $90 \mu s$ of equilibrium HPS model[47] simulation with one TDP-43 LCD and one CK1$\delta$.

The neural network returns the probability of being in one of the 2 output states (ideally bound and unbound state) for each snapshot of the trajectory. We assigned each snapshot to the state with higher probability, filtering those with a probability between 30% and 70% using transition-based state assignment[87]. In other words, these configurations were assigned based on the state of the previous and following snapshots, in order to filter out spurious transitions.

In order to test the generality of our method, we repeated the bound state recognition withVAMPnet, but using more input nodes. In particular, we used the distances between all the residues of TDP-43 LCD and 30 equally spaced residues of CK1$\delta$, resulting in an input layer of 4620 nodes. This time we used 2 hidden layers with 154 and 30 nodes each, and an output layer with 2 nodes. We reduced the learning rate to $0.5 \times 10^{-3}$ and the batch size to $10^4$.

For the VAMPnet analysis, the Python packages Deeptime version 0.4.4[67], PyTorch version 2.0.0 and PyEMMA version 2.5.12[88] have been used. The code for the VAMPnet analysis is available in the Zenodo repository[83].

## Implied timescales and Chapman-Kolmogorov test

The choice of the lag time was done by looking at the implied timescales. We can estimate the implied timescales of the Markov model from the eigenvalues of the transition matrix as:

$$t_i(\tau) = -\frac{\tau}{\ln |\lambda_i(\tau)|} \tag{5}$$

with $\lambda_i(\tau)$ the eigenvalues of $T_{ij}(\tau)$. We chose a lag time $\tau$ such that $t_i(\hat{\tau})$ is approximately constant for every $\hat{\tau} \geq \tau$. We estimated $\Delta\mu_{cycle}$ for different lag times (1, 10, 20 Markov chain steps) for the case of reactive Ser 403 and $\Delta\mu_P = 0, -5 \, \text{kJ mol}^{-1}$. For $\tau \geq 10$ Markov chain steps, the estimated $\Delta\mu_{cycle}$ is in agreement with $\Delta\mu_P$ (Supplementary Table 2).

We estimated the goodness of the MSM by looking at the Chapman-Kolmogorov test (CK test)[59,69]. In a Markovian process, the transition matrix satisfies the relation

$$T_{ij}(n\tau) = [T_{ij}(\tau)]^n \tag{6}$$

with $n \geq 1$. In other words, the transition matrix of the model estimated at lag time $n\tau$ must be equal to the transition matrix to the power $n$ of the model estimated at lag time $\tau$. The CK test compares $T_{ij}(n\tau)$ (the estimated transition matrix) and $T_{ij}^n(\tau)$ (the predicted transition matrix) for every possible transition $i \rightleftharpoons j$ and different lag times $n\tau$.

## Estimation of phosphorylation rates

In all the single TDP-43 LCD chain simulations, we estimated the phosphorylation rates $r_P$ assuming the phosphorylation process is without memory and thus follows single-exponential kinetics. In all the

collected $N_{sim} = 100$ simulations, we had at most one phosphorylation event for each Ser residue, happening at time $t_P^i < t_{tot}$ for simulation $i$, with $t_{tot}$ the total time of the simulation. In this case, we can use the maximum likelihood estimator for the rate with a uniform prior distribution[74]

$$r_P = \frac{n+1}{\Theta}, \quad \text{var}(r_P) = \frac{n+1}{\Theta^2}, \text{with} \quad \Theta = \sum_{i=1}^n t_P^i + (N_{sim} - n)t_{tot} \tag{7}$$

where $n$ is the counts of simulations with one phosphorylation event for the Ser took in consideration. Instead, for the simulations in condensate, in which we have multiple TDP-43 chains and thus multiple phosphorylation events for each Ser residue, we computed $r_P$ as the total count of phosphorylation events in the simulation divided by the total simulation time. In this case, the error on the estimate of the rate is computed as the standard error of the mean from the different replicas. In the same way, we also computed all the contacts rates $r_c$ and their error.

However, since the phosphorylation of a Ser can happen only if TDP-43 is bound to the enzyme, it is more appropriate to take into account the conditional probability of the phosphorylation event given the binding of TDP-43 and CK1$\delta$ already occurred. Given $p_B(t)dt = r_B \exp(-r_B t)dt$ the probability of binding between time $t$ and $t + dt$ and $p_P(t)dt = r_P \exp(-r_P t)dt$ the probability of having a phosphorylation between time $t$ and $t + dt$, the conditional probability of having a phosphorylation between time $t$ and $t + dt$ given that TDP-43 is bound to CK1$\delta$ is

$$P(t|\text{bound})dt = \int_0^t p_B(t')dt' p_P(t - t')dt = \frac{r_B r_P}{r_B - r_P}\left(e^{-r_P t} - e^{-r_B t}\right)dt \tag{8}$$

If we call $P_c(t < T)$ the probability of having a phosphorylation event within time $T$ in our simulations, we can write its complementary as:

$$1 - P_c(t < T) = 1 - \int_0^T P(t)dt = \frac{r_B r_P}{r_B - r_P}\left(\frac{e^{-r_P t}}{r_P} - \frac{e^{-r_B t}}{r_B}\right) \tag{9}$$

Instead, if we assume that the binding process is much faster than the phosphorylation one ($r_B \gg r_P$), than we can approximate $P(t|\text{bound}) \sim p_P(t)$ and $1 - P_c(t < T) \sim \exp(-r_P t)$.

From the 100 simulations used to estimate the phosphorylation rates, we computed the normalized inverse cumulative histogram of the phosphorylation events time $T$, where each time bin gives the phosphorylation counts for that bin plus the counts of all the following bins, divided by the number of simulations. We fitted it with $1 - P_c(t < T)$ both for a single-exponential process and a conditioned single-exponential process.

## Condensate identification with clustering analysis

In order to identify the TDP-43 LCD condensate in the trajectory file, we used the DBSCAN (Density-Based Spatial Clustering of Applications with Noise) clustering analysis algorithm[89] available in the Python package scikit-learn (version 1.2.2 has been used). The code is available in the Zenodo repository[83].

DBSCAN is an efficient algorithm to identify clusters based on an Euclidean distance cut-off $\epsilon$ and a minimum cluster size parameter $n_{min}$. Particles with at least $n_{min}$ neighbors within a distance $\epsilon$ are considered core particles of the cluster. Instead, particles with fewer than $n_{min}$ neighbors are considered non-core particles and they are assigned to a cluster only if at least one of their neighbors is a core particle.

For the estimate of the condensate size and of the percentage of CK1$\delta$ in contact with the condensate, we used the positions of every bead as input data, and we chose the parameters $\epsilon = 1 \, \text{nm}$ and $n_{min} = 2$.

With this choice, every isolated chain is considered a cluster and two different chains belong to the same cluster whenever at least one of their particles is in contact (within 1 nm). However, varying $\epsilon$ between 0.8 nm and 3 nm and $n_{min}$ between 2 and 5 does not significantly change the results. We accounted for the periodic boundary conditions by centering the condensate in the box at every frame.

## Reporting summary

Further information on research design is available in the Nature Portfolio Reporting Summary linked to this article.

## Data availability

Example trajectories and code for analysis are available as a Zenodo repository[83] (https://doi.org/10.5281/zenodo.13833525). Source data are provided in this paper.

## Code availability

The code used for the simulations and the HOOMD-blue plugin with the Ashbaugh-Hatch pair potential for the non-bonded interactions are available on GitHub (https://github.com/ezippo/hoomd3_phosphorylation, https://github.com/ezippo/ashbaugh_plugin)and Zenodo[83,85,86] (https://doi.org/10.5281/zenodo.13833525, https://doi.org/10.5281/zenodo.15207448, https://doi.org/10.5281/zenodo.15207491).

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

## Acknowledgements

E.Z. Funded by the Deutsche Forschungsgemeinschaft (DFG, German Research Foundation) - Project number 233630050 - TRR 146. L.S.S. thanks M$^3$ODEL and ReALity (Resilience, Adaptation and Longevity) and Forschungsinitiative des Landes Rheinland-Pfalz for support. This project was funded by the Deutsche Forschungsgemeinschaft (DFG, German Research Foundation) - SFB 1551 - Project No. 464588647. Further, we gratefully acknowledge the computing time granted on the supercomputers Mogon II at Johannes Gutenberg University Mainz, which is a member of the AHRP (Alliance for High Performance Computing in Rhineland Palatinate) and the Gauss Alliance e.V. We thank Alex Holehouse, Heinz Köppl, Martin Girard, and Friederike Schmid, for insightful discussions.

## Author contributions

E.Z. ran the simulations, analyzed data, interpreted results, wrote the manuscript. D.D. provided important intellectual knowledge and assisted in designing the study, reviewed the manuscript. T.S. conceived the study, provided important intellectual knowledge, reviewed the manuscript. L.S.S. conceived and supervised the study, interpreted results, wrote, and reviewed the manuscript.

## Funding

## Competing interests

The authors declare no competing interests.
