## [Transparent Peer Review file · Nature Communications]

Molecular simulations of enzymatic phosphorylation of disordered proteins and their condensates

Corresponding Author: Professor Lukas Stelzl

Version 0:

Reviewer comments:

Reviewer #1

(Remarks to the Author)

In this paper, Zippo et al. present a method that combines Monte Carlo (MC) and molecular dynamics (MD) simulations to study the phase behavior of proteins involving chemical reactions. The driving energy from ATP hydrolysis is incorporated into the enzymatic phosphorylation cycle of TDP43, whose thermodynamic consistency is validated using Markov state models. Through simulations that allow multiple phosphorylations, the authors attempt to determine the reasons for the differences in phosphorylation rates between N-terminal and C-terminal serine residues. By extending their method, they also simulate the phosphorylation-coupled dissolution of TDP43 droplets. The effect of the disordered tail in the enzyme is also discussed.

Overall, this article is detailed and comprehensive, with reliable simulations and data analysis. But, I recommend the author to reply several concerns or comments in their revised manuscript.

Major concerns

1. The description of the MC component is not sufficiently clear. For example, if I understand correctly, $\Delta\mu_P$ controls only the relative magnitudes of k_{23} and k_{32} (or the transition probabilities T_{23} and T_{32}). On what basis are the absolute values of these quantities set? How do their absolute values affect the entire MC + MD cycle? Additionally, in Figure 1, there are arrows with different lengths and thicknesses. Do they signify different meanings? For instance, the arrow representing k_{23} is thicker than the others; does this indicate that k_{23} is significantly larger than the others?
2. It is surprising that the "averaged interaction" leads to a difference of an order of magnitude in contact probability compared to the wild type. According to the description in Line 496, I expected to find the averaged values of the parameters in the Supporting Information, but they are absent. The authors should provide a direct description of these parameters in the main text.
3. The last paragraph on Page 7 suggests that the positive charges in the N-terminus determine the differences in phosphorylation rates between the N- and C-termini. However, the first paragraph on Page 11 indicates that even without considering the specific charge distribution, the C-terminal serine residues still exhibit higher phosphorylation rates. Are these two statements contradictory? Which factor plays a more decisive role in determining the phosphorylation rate—the charge pattern or the serine sequence pattern?
4. It is interesting to observe that in simulations with a single enzyme and a single TDP43 molecule, binding and unbinding processes occur, whereas in droplet simulations, the enzyme (without the disordered tail) mainly binds on the surface of the condensate (Figure 4). In this case, is the primary contribution to the interaction between the enzyme and TDP43 from LJ or electrostatic interactions? Why does the enzyme remain on the surface rather than entering the interior of the TDP43 condensate?
5. After phosphorylation, does TDP43 redistribute within the condensate? Do phosphorylated sites remain on the surface? Does this affect the "saturation" of the phosphorylation rate? The authors mention that the overall phosphorylation percentage starts to induce dissolution when it reaches 24%. For each individual chain, at what phosphorylation percentage does it migrate into the dilute phase?
6. The authors found that enzymes mainly bind to the surface of the condensates, even when the migration of phosphorylated TDP43 from the condensate was observed (Figure 4). However, it was also claimed that phosphorylation increases the binding affinity between TDP43 and the enzyme (Page 10). In this sense, shouldn't the enzyme be more strongly attracted to phosphorylated TDP43 in the dilute phase?
7. While ATP and ADP are modeled implicitly, the concentrations of these molecules may also affect phase separation (D Kota, ..., HX Zhou. 2024. JACS 146, 1326; M Dang, ..., J Song. 2021. Commun Biol 4, 714). The authors should discuss this

point to provide a more comprehensive understanding.

Minor points

1. Figure 2c: The description of the colored dots and lines is missing. Please include a legend or description to clarify what they represent.
2. Figure 2e: Change "kJ\mol" to "kJ/mol". Additionally, it would be better if both axes have the same precision for consistency.
3. Line 143: Consider changing "goodness" to "reliability" to enhance clarity.
4. Figure 3f: The thickness of the arrows pointing from "np" to "pS410" indicates the percentage. How about the arrows connecting "np" and "(n+1)p"? If they have a different meaning, consider using a different drawing style. Also, please include a detailed explanation in the figure caption.

(Remarks on code availability)

Reviewer #2

(Remarks to the Author)

This paper explores the molecular mechanisms by which phosphorylation, a key regulatory modification, influences the behavior of intrinsically disordered proteins (IDPs) and their phase-separated condensates. Using molecular dynamics (MD) simulations, the authors simulate enzymatic phosphorylation events and analyze the subsequent changes in IDP structure, intramolecular interactions, and condensate formation. The study specifically focuses on biomolecular condensates formed by TDP-43, a protein linked to neurodegenerative disease, thus addressing an important biological question. The authors' findings provide insights into how phosphorylation modulates the stability and material properties of these condensates, advancing our understanding of post-translational regulation in biomolecular phase separation.

Overall, this is a very strong paper and appears suitable for publication in its current form.

Questions for the Authors:

1. Could the authors comment on how phosphorylation patterns found in patient samples may be mechanistically linked to the disruption of condensate formation? Given that understanding the pathological role of phosphorylation in TDP-43 is central to this work, a brief discussion—even speculative—would help frame the broader implications of these findings.
2. There have been recent reports correlating TDP-43 phase separation with functional aspects of the protein (e.g., bioRxiv, DOI: 10.1101/2024.07.05.602258). Can the authors clarify whether similar functionality-related effects of phase separation are being modeled here, or is this system reflecting a more complex behavior specific to TDP-43?
3. While the model seems suitable for studying IDP phase behavior, it is less clear if it accurately represents interactions between TDP-43's C-terminal domain and CK1δ. Could the authors discuss any limitations of their model in capturing these specific binding events?
4. As it is well known that TDP-43 CTD phase separation is driven by interactions mediated by its conserved helical region, the authors should comment on how they model this helical region. Additionally, how does the helical region factor into interactions with CK1δ and the resulting phosphorylation patterns?

(Remarks on code availability)

Reviewer #3

(Remarks to the Author)

(Remarks on code availability)

Version 1:

Reviewer comments:

Reviewer #1

(Remarks to the Author)

The authors satisfactorily revised the manuscript according to the reviewers' comments on the original manuscript.

(Remarks on code availability)

Reviewer #2

(Remarks to the Author)

The manuscript has significantly improved, with many details added to the main text and supporting information. The authors have addressed our comments well; however, a few minor points remain:

1. We thank the authors for incorporating a new set of simulation results to account for the effects of the conserved helix region by fixing its helical structure using a rigid body approximation. However, the rationale behind arbitrarily scaling down hydrophobic interactions within this region (by 30%), as they do for folded domains, remains unclear. As initially proposed by Dignon et al. (PMID: 29364893), scaling down interactions among folded domains restrained as rigid bodies could be reasonable due to the presence of buried residues. However, it is unclear how this same principle applies to TDP-43 CR helical structure, which does not contain buried residues. In principle, helix–helix interactions should be significantly stronger than those of the flanking domains, as supported by recent experimental findings (Mohanty et al. 2023, PMID: 37579155). That being said, it is worth noting that single-bead-per-residue CG models cannot fully capture the complex nature of site-specific interactions by TDP-43 CR. In particular, these models do not account for how its partial helicity stabilizes and extends across the entire CR upon intermolecular helix–helix interactions, nor do they capture sequence-specific changes to its helical structure that may occur due to phosphorylation of serines within the region, which also modulates its phase separation (Haider et al. 2024, PMID: 38178578).

2. An additional point for the authors to consider is the need for a clearer rationale for choosing the HPS-cation- π model. Furthermore, while cation- π interactions are discussed in several instances, it may be beneficial to acknowledge that the drivers of TDP-43 CTD phase separation extend beyond π - π and cation- π interactions. Mohanty et al. (PMID: 37579155) demonstrated that methionine residues, both within and outside the CR, play a crucial role in phase separation. Notably, methionine residues within the CR have a significantly greater impact, suggesting that phase separation is strongly enhanced through helix–helix contacts stabilized by methionine residues in this region. The authors may decide whether and how to incorporate these points, but acknowledging them could provide a more comprehensive perspective on the molecular basis of TDP-43 CTD phase separation.

3. To improve clarity for readers, it may be helpful to include a sequence analysis of CK1 δ , highlighting the surface-exposed residues, active site, and its disordered tail.

4. The authors mention that when they switch off the arbitrary rescaling of hydrophobic interactions for CK1 δ , the enzyme localizes to the interior of the condensate. Could they comment on how this scenario impacts the results compared to the case where CK1 δ is positioned on the surface of the condensate?

As this is the second round of reviews, I leave it to author's discretion whether to address these points or not.

(Remarks on code availability)

Reviewer #3

(Remarks to the Author)

(Remarks on code availability)

Reviews reply

Emanuele Zippo, Dorothee Dormann, Thomas Speck, Lukas Stelzl

Review 1

In this paper, Zippo et al. present a method that combines Monte Carlo (MC) and molecular dynamics (MD) simulations to study the phase behavior of proteins involving chemical reactions. The driving energy from ATP hydrolysis is incorporated into the enzymatic phosphorylation cycle of TDP43, whose thermodynamic consistency is validated using Markov state models. Through simulations that allow multiple phosphorylations, the authors attempt to determine the reasons for the differences in phosphorylation rates between N-terminal and C-terminal serine residues. By extending their method, they also simulate the phosphorylation-coupled dissolution of TDP43 droplets. The effect of the disordered tail in the enzyme is also discussed. Overall, this article is detailed and comprehensive, with reliable simulations and data analysis. But, I recommend the author to reply several concerns or comments in their revised manuscript.

We thank the reviewer for their positive assessment of our simulations and the very thoughtful comments, which helped us to improve our manuscript further.

1. The description of the MC component is not sufficiently clear. For example, if I understand correctly, $\Delta\mu_P$ controls only the relative magnitudes of k_{23} and k_{32} (or the transition probabilities T_{23} and T_{32}). On what basis are the absolute values of these quantities set? How do their absolute values affect the entire MC + MD cycle? Additionally, in Figure 1, there are arrows with different lengths and thicknesses. Do they signify different meanings? For instance, the arrow representing k_{23} is thicker than the others; does this indicate that k_{23} is significantly larger than the others?

We apologize for the lack of clarity. We agree on the necessity of a more extensive analysis and clarification of the cycle kinetics. In order to clarify the role of $\Delta\mu_P$ in the cycle kinetics, we wrote in the main text “*The transitions $1 \rightleftharpoons 2$, $3 \rightleftharpoons 4$ (the binding/unbinding of the enzyme with TDP-43 or phosphorylated TDP-43) and $4 \rightleftharpoons 1$ (the reservoir exchange step) are not driven by external energy sources, thus we expect the ratio of rates k_{ij}/k_{ji} to be independent from $\Delta\mu_P$ and be determined by the equilibrium free energies of the states. By contrast, the phosphorylation MC step $2 \rightleftharpoons 3$ is driven by $\Delta\mu_P$, which creates a net probability current (flowing clockwise in the sketch of Fig. 1 if $\Delta\mu_P < 0$) that breaks the detailed balance condition (Knoch New J. Phys. 2015). We then expect k_{23}/k_{32} to be determined not only by the equilibrium free energies, but also by $\Delta\mu_P$.*” We show in new Supplementary Fig. 2 that the logarithm of the transition probabilities ratio k_{23}/k_{32} grows linearly with $\Delta\mu_P$, while the other ratios are independent from $\Delta\mu_P$.

We also clarify the effect of the absolute value of T_{ij} on the cycle in “*It is interesting to observe that $\Delta\mu_{\text{cycle}}$ is exclusively determined by the ratio between forward and backward transition probabilities, while their absolute value is not relevant by itself.*” On the other hand, we show in new Supplementary Fig. 1 how the absolute value of the transition rates change with the attempt rate of the MC move. We added in the text “*We note that the absolute values of k_{23} and k_{32} depend also on the attempt rate of the MC step (Methods), which though does not affect their ratio (Supplementary Fig. 1, Supplementary Text).*” Additional clarifications are added in the section “Detailed balance breaking” of the Supplementary Text.

We agree that the difference in the thickness of the arrow in Fig. 1 could be misleading, for

this reason we represented all the arrows in Fig. 1 with same thickness and added in Fig. 2c an illustration of a cycle with arrow sizes proportional to the rates. The original idea was to emphasize that the cycle kinetics is driven by the high concentration of ATP, which pushes the phosphorylation reaction out of equilibrium.

2. It is surprising that the "averaged interaction" leads to a difference of an order of magnitude in contact probability compared to the wild type. According to the description in Line 496, I expected to find the averaged values of the parameters in the Supporting Information, but they are absent. The authors should provide a direct description of these parameters in the main text.

We thank the reviewer for pointing out a lack of explanation regarding the difference of an order of magnitude in the phosphorylation rates between wild type TDP-43 and average interaction strength chain. The main reason for this difference might be due to the absence of charges and aromatic residues in the averaged chain. We added in the text "*The averaged interaction strength bead has mass, size and hydrophobicity averaged from the wild type TDP-43 LCD, while the electric charge is zero (Methods). Note that there are no cation- π interactions in this case. Due to the absence of aromatic residues and cation- π interactions, the phosphorylation rates (Fig. 3b,h), as well as the contact frequency in equilibrium (Fig. 3c,i), are one order of magnitude smaller in the case of averaged interaction sequence compared to the wild type TDP-43.*" We added the exact values of the average parameters in the Methods section: "*The averaged interaction polymer is built by substituting the TDP-43 LCD residues different from Ser with a bead having zero electric charge and average TDP-43 LCD mass (98.957 amu), size parameter σ (0.54331 nm) and hydrophathy parameter λ (0.64039)*". In order to clarify the structure of the 2 types of averaged interaction chains, we added in Fig. 3g a sketch of the position of the Ser residues along the chain.

3. The last paragraph on Page 7 suggests that the positive charges in the N-terminus determine the differences in phosphorylation rates between the N- and C-termini. However, the first paragraph on Page 11 indicates that even without considering the specific charge distribution, the C-terminal serine residues still exhibit higher phosphorylation rates. Are these two statements contradictory? Which factor plays a more decisive role in determining the phosphorylation rate—the charge pattern or the serine sequence pattern?

To better understand the the role of charges in TDP-43 phosphorylation, we ran simulations of TDP-43 with the charges of residues different from phosphoserines switched off. We report the results in Fig. 3b-c and new Supplementary Fig. 13-14. The Reviewer is right and we corrected our analysis by adding in the text "*We found that the contact frequency r_c without phosphorylations is higher for the N-terminal Ser residues (compared to wild type TDP-43), while for the C-terminal we get comparable results, with some Ser residues showing a small increase in the number of contacts (Fig. 3c, in red). However, the results in Fig. 3b (in red) show a decrease of phosphorylation rates r_p by roughly a factor of 4 for the N-terminal phosphosites and for the C-terminal Ser residues in the proximity of aromatic residues (Ser 369, Ser 387, Ser 393, Ser 395, Ser 403 and Ser 410), while the phosphorylations of the other C-terminal Ser are even more suppressed. This means that, even though the positive charges in the N-terminus screen the interaction with the active site of CK1 δ in absence of pSer, they help the phosphorylation when some serines are already phosphorylated.*". We also added in the Discussion "*We also highlight how both serine distribution and presence of charges enhance the phosphorylation of C-terminal Ser residues (Fig. 3b,h,g). However, aromatic residues, and in general interaction-prone residues, in the C-terminus (Yan bioRxiv 2024) favor the formation of contacts with the active site of the kinase even in simulations of TDP-43 LCD without charges and phosphorylated serines (Fig. 2b), which shows once more the importance of the sequence context.*"

4. It is interesting to observe that in simulations with a single enzyme and a single TDP43 molecule, binding and unbinding processes occur, whereas in droplet simulations, the enzyme

(without the disordered tail) mainly binds on the surface of the condensate (Figure 4). In this case, is the primary contribution to the interaction between the enzyme and TDP43 from LJ or electrostatic interactions? Why does the enzyme remain on the surface rather than entering the interior of the TDP43 condensate?

Binding and unbinding processes occur without phosphorylations both in the single TDP-43 simulations and in condensate. Supplementary Fig. 17 right shows that in average only 40% of the enzyme chains are attached to the condensate in equilibrium MD simulations of unphosphorylated TDP-43 LCD. Also, in both cases phosphorylations make the enzyme to interact more strongly with TDP-43, thus unbinding events become rare (Fig. 4d).

We computed the energy contributions from each pair potential for every residue of CK1 δ from snapshots with enzyme bound to the condensate with 0% and 40% of phosphorylated serines (new Supplementary Fig. 21). We added to the text “*The hydrophobic interactions are in general the dominant ones across the enzyme chain, with average strength per residue of (-0.117 ± 0.006) kJ mol $^{-1}$ in the case without pSer and (-0.209 ± 0.008) kJ mol $^{-1}$ in the case with 40% pSer. The cation- π interactions are also important, even though more localized, reaching strengths lower than -0.4 kJ mol $^{-1}$ for some residues. However, the most relevant difference is in the electrostatic interactions, with positively charged residues going from a mean positive value of the potential energy in absence of pSer, which oppose the cation- π interactions, to strong negative values for the case with 40% pSer (the opposite happens to the negatively charged residues). This reflects in the average strength per residue, going from $(+0.006 \pm 0.003)$ kJ mol $^{-1}$ without pSer to (-0.17 ± 0.06) kJ mol $^{-1}$ with 40% pSer.”*

Moreover, the lower hydrophobicity of CK1 δ compared to TDP-43 LCD in our model seems to explain why the enzyme remains on the surface of the condensate. The CK1 δ folded domain is modeled as a rigid body and its hydrophobic interactions are scaled down by 30%. By removing the rescaling of the folded domain, we saw the enzyme entering the interior of the condensate. We wrote “*The reason why CK1 δ chains locate on the surface of the condensate might be linked to the higher hydrophobicity of TDP-43 LCD. Indeed, if we switch off the rescaling of the hydrophobic interactions (Methods) for CK1 δ , we observe that the enzyme locates in the interior of the condensate (Supplementary Fig. 18).*”

5. After phosphorylation, does TDP43 redistribute within the condensate? Do phosphorylated sites remain on the surface? Does this affect the “saturation” of the phosphorylation rate? The authors mention that the overall phosphorylation percentage starts to induce dissolution when it reaches 24%. For each individual chain, at what phosphorylation percentage does it migrate into the dilute phase?

We computed the radial density profile of Ser and pSer and CK1 δ residues in condensate from simulations in equilibrium with 40% of phosphorylated serines (new Fig. 4c). We added in the text “*... the action of the kinases is limited to the surface of the droplet and the density of pSer residues in the condensate remains higher at the interface, supporting the idea of an early saturation of the most accessible phosphosites. [...] Most CK1 δ residues are located at the interface of the condensate (at around 11 nm from the center). By looking at the inset (of Fig. 4c) in semi-log scale, we notice that the density of pSer is higher than the Ser one at distances greater than 14 nm, reaching 25% of the total number of beads in the bins for $r \geq 18$ nm (Supplementary Fig. 17 left). Indeed, the hyperphosphorylated chains often stretch out of the condensate before leaving it, as visible in Fig. 4a at 3×10^8 MD steps.*”

In order to show at which phosphorylation percentage each chain leave the condensate, we wrote “*It is also interesting to observe that in our simulations TDP-43 chains mostly leave the condensate when an average of 16 Ser residues (out of 24) are phosphorylated within the same chain, as shown in Supplementary Fig. 20, where we report the distribution of the number of pSer per chain in dilute phase at different times through a violin plot.*”

6. The authors found that enzymes mainly bind to the surface of the condensates, even when the migration of phosphorylated TDP43 from the condensate was observed (Figure 4). However, it was also claimed that phosphorylation increases the binding affinity between TDP43

and the enzyme (Page 10). In this sense, shouldn't the enzyme be more strongly attracted to phosphorylated TDP43 in the dilute phase?

Even in the case in which there are some hyperphosphorylated chains in dilute phase, the density of phosphorylated chains is higher in the condensate. Moreover, the enzymes locate at the interface, where the relative density of pSer residues is even higher than the Ser residues one (new Fig. 4c, new Supplementary Fig. 17 left). We also ran simulations of the condensate with 40% of the total Ser residues being phosphorylated with a CK1 δ placed in dilute phase. In all replicas, after transiently binding to some hyperphosphorylated TDP-43 chains in dilute phase, the enzyme binds to the condensate (Supplementary Fig. 19). We added in the text “*Even in equilibrium simulations at 40% pSer with 1 CK1 δ chain placed outside of the condensate, we observe that the enzyme binds to the condensate rather than remaining bound to the hyperphosphorylated chains in dilute phase (Supplementary Fig. 19, Movie 6). This behavior might be due to the higher density of pSer on the surface of the condensate than in the dilute phase (Fig. 4c).*”

7. While ATP and ADP are modeled implicitly, the concentrations of these molecules may also affect phase separation. The authors should discuss this point to provide a more comprehensive understanding.

We write in Discussion: *In our simulations ATP and ADP are modeled implicitly, but the interactions of ATP can modulate interactions of proteins in condensates (Kota JACS 2024, Dang Comm Biol 2023) and one can envisage to explicitly simulate ATP and ADP to capture such effects in coarse-grained molecular dynamics simulations.*

Minor points

1. Figure 2c: The description of the colored dots and lines is missing. Please include a legend or description to clarify what they represent.
2. Figure 2e: Change "kj/mol" to "kJ/mol". Additionally, it would be better if both axes have the same precision for consistency.
3. Line 143: Consider changing "goodness" to "reliability" to enhance clarity.
4. Figure 3f: The thickness of the arrows pointing from "np" to "pS410" indicates the percentage. How about the arrows connecting "np" and "(n+1)p"? If they have a different meaning, consider using a different drawing style. Also, please include a detailed explanation in the figure caption.

We thank the reviewer for the corrections and suggestions. All the minor points were taken into consideration and the text was modified accordingly.

Review 2

This paper explores the molecular mechanisms by which phosphorylation, a key regulatory modification, influences the behavior of intrinsically disordered proteins (IDPs) and their phase-separated condensates. Using molecular dynamics (MD) simulations, the authors simulate enzymatic phosphorylation events and analyze the subsequent changes in IDP structure, intramolecular interactions, and condensate formation. The study specifically focuses on biomolecular condensates formed by TDP-43, a protein linked to neurodegenerative disease, thus addressing an important biological question. The authors' findings provide insights into how phosphorylation modulates the stability and material properties of these condensates, advancing our understanding of post-translational regulation in biomolecular phase separation.

Overall, this is a very strong paper and appears suitable for publication in its current form.

We thank the reviewer for their positive and encouraging assessment.

1. Could the authors comment on how phosphorylation patterns found in patient samples may be mechanistically linked to the disruption of condensate formation? Given that understanding the pathological role of phosphorylation in TDP-43 is central to this work, a brief discussion—even speculative—would help frame the broader implications of these findings.

In Discussion we write:

“Our simulations suggest that enzymatic phosphorylation of TDP-43 by CK1δ can occur in condensates, which is a step towards elucidating the molecular consequences of this PTM and in particular phosphorylation patterns detected in patient samples. Experiments will be required to test whether enzymatic phosphorylation of TDP-43 in condensates is dominant mode of how phosphorylation dissolves TDP-43 condensates and may protect against the formation of insoluble TDP-43 aggregates. This may be a key mechanism in the early stages of the formation of aggregates in neurons (Sternburg Trends Biochem Sci 2022). TDP-43 hyperphosphorylation was previously shown to reduce TDP-43 condensation and aggregation (Grujis da Silva EMBO J 2022), but whether enzymes phosphorylated TDP-43 in dilute solution or in the condensates was not know. Our simulations show that partially phosphorylated TDP-43 condensates are stable. One can speculate that TDP-43 phosphorylation might change how TDP-43 and TDP-43 condensates in the nucleus interact with their cognate binding partners (Hallegger Cell 2021, Aikio CellRep 2025, Sternburg Trends Biochem Sci 2022) and how aberrant TDP-43 condensates in the cytoplasm interact with cytoplasmic proteins (Yan bioRxiv 2024), which could lead to toxic gain-of function or loss-of-function (Wang EMBO J 2018).

TDP-43 phosphorylation could also reduce its nuclear localization and induce its accumulation in the cytoplasm, which is associated with neurodegenerative disease (Grujis da Silva EMBO J 2022, Yan bioRxiv 2024). Recently, it was demonstrated that mutations in the C-terminal region of TDP-43 which reduce the stability of the C-terminal helix (residues 319-341) also reduce nuclear localization of TDP-43 (Rizuan bioRxiv 2024). Hence, the C-terminal phosphorylations (e.g., Ser 332) we investigated could trigger the cytoplasmic localization of TDP-43 by disrupting structure of the C-terminal helix, as has been shown in biophysical studies of the TDP-43 LCD (Haider Biophys J 2024, Conicella PNAS 2020, Conicella Structure 2016).”

2. There have been recent reports correlating TDP-43 phase separation with functional aspects of the protein. Can the authors clarify whether similar functionality-related effects of phase separation are being modeled here, or is this system reflecting a more complex behavior specific to TDP-43? bioRxiv, DOI: 10.1101/2024.07.05.602258

We thank the reviewers for prompting us to link our simulation to functionality-related effects of phase separation. We now write in the Introduction *“TDP-43 condensates are functionally important (Hallegger Cell 2021, Wang EMBOJ 2018, Sternburg Trends Biochem Sci 2022). The LCD is critical for the functional roles of TDP-43 in, e.g., splicing and 3' polyadenylation (Wang EMBO J 2018, Hallegger Cell 2021, Conicella PNAS 2020). A conserved helix (residues 319-341) within the LCD has been shown to be important for phase separation*

(*Conicella Structure 2016, Rizuan bioRxiv 2024*) and the physiological roles of TDP-43 in RNA processing (*Conicella PNAS 2020, Hallegger Cell 2021*). In diseased neurons, TDP-43 loses its nuclear localization and forms aggregates in the cytoplasm". In the Discussion we refer to the recent results by Rizuan bioRxiv, DOI: 10.1101/2024.07.05.602258 on how a de-stabilization of the C-terminal helix might have a role beyond disruption of phase separation by favoring the cytoplasmic (linked to disease) over the nuclear localization of TDP-43 (in homeostasis).

3. While the model seems suitable for studying IDP phase behavior, it is less clear if it accurately represents interactions between TDP-43's C-terminal domain and CK1 δ . Could the authors discuss any limitations of their model in capturing these specific binding events?

As a check we repeated our investigation of the TDP-43 C-terminal domain and CK1 δ with the recently developed CALVADOS3 simulation model, which has been parameterized to capture the interactions of folded domains and disordered regions.

In Discussion we write: *It is important to note that details of the conformations of proteins will be critical for the molecular recognition of potential phosphorylation sites by kinases and more detailed and all-atom molecular simulations (Zhang JACS 2024, Rizuan bioRxiv 2024) and high-resolution experiments (Zhang JACS 2024, Rizuan bioRxiv 2024) will be required to fully understand the recognition mechanisms. However, the comparison with CALVADOS3 force field (Supplementary Fig. 15), which employs elastic network rather than rigid body dynamics for the folded domains, is encouraging and suggest of that coarse-grained models capture -at least in part- sequence specific interactions.*

4. As it is well known that TDP-43 CTD phase separation is driven by interactions mediated by its conserved helical region, the authors should comment on how they model this helical region. Additionally, how does the helical region factor into interactions with CK1 δ and the resulting phosphorylation patterns?

In the initial simulations, we did not impose a 3D structure on the helix. We reasoned that the residue-level coarse-grained force field might partially capture the enhanced interaction propensity of this conserved region in an effective way, considering that it is part of the "hydrophobic path" (Yan bioRxiv 2024). We add in Results:

"In our simulations, the interactions of the helical region of TDP-43 (Rizuan bioRxiv 2024, Conicella PNAS 2020, Conicella Structure 2016) are determined by the interaction parameters of the residues of this region, which is enriched in hydrophobic residues (Hallegger Cell 2021, Yan bioRxiv 2024)."

To more directly address the role of structure of the C-terminal helix we investigated how explicitly modeling the helix in the dilute and dense phase affects our simulations:

"Another concern is that our simulations might underestimate or overestimate interactions of the conserved region (residues 319-341) of TDP-43 LCD, which adopts helical conformations. In our initial simulations the effect of the helix was captured only in an effective sense by, through the simulation parameters of the residues in this region. Nonetheless, explicitly modeling the C-terminal helix by fixing residues 320-332 as rigid body (Supplementary Text), reduces how often Ser 332 in the helix is in contact with the CK1 δ active site, but otherwise does not affect the contact statistics of TDP-43 LCD Ser residues with the CK1 δ active site (Supplementary Fig. 8)."

"As for simulations in the dilute phase (Supplementary Fig. 8, we also checked that we obtained similar results by modeling the helical structure in residues 319-341 explicitly (Supplementary Fig. 23, Supplementary Text). Dissolution of the TDP-43 condensates in simulations with 5 CK1 δ chains starts when about 26% of Ser are phosphorylated, Supplementary Fig. 23), which is similar to simulations without enforcing the helical structure."

Reviewer 3 (Remarks to the Author)

We thank the reviewer for helping us to improve our manuscript by contributing to the listed reports. Thank you!

REVIEWERS' COMMENTS

Reviewer #1 (Remarks to the Author):

The authors satisfactorily revised the manuscript according to the reviewers' comments on the original manuscript.

We are pleased to hear that the revised manuscript addresses the reviewers' comments in satisfactory way.

Reviewer #2 (Remarks to the Author):

The manuscript has significantly improved, with many details added to the main text and supporting information. The authors have addressed our comments well; however, a few minor points remain:

We thank the reviewer for their positive and encouraging assessment.

1. We thank the authors for incorporating a new set of simulation results to account for the effects of the conserved helix region by fixing its helical structure using a rigid body approximation. However, the rationale behind arbitrarily scaling down hydrophobic interactions within this region (by 30%), as they do for folded domains, remains unclear. As initially proposed by Dignon et al. (PMID: 29364893), scaling down interactions among folded domains restrained as rigid bodies could be reasonable due to the presence of buried residues. However, it is unclear how this same principle applies to TDP-43 CR helical structure, which does not contain buried residues. In principle, helix–helix interactions should be significantly stronger than those of the flanking domains, as supported by recent experimental findings (Mohanty et al. 2023, PMID: 37579155).

We thank the reviewer for this excellent point. In response, we have repeated the simulations with the helix without the rescaling of the hydrophobic interactions. We now write in the Results:

“Explicitly modeling the C-terminal helix by fixing residues 320-332 as rigid body (Supplementary Methods), increases contacts of Ser residues in this region with the CK1δ active site, most notably Ser 317, Ser 332, and Ser 333 (Supplementary Fig.9), but otherwise does not affect the contact statistics of TDP-43 LCD Ser residues with the CK1δ active site. Besides the helical region Ser residues, in the C-terminal part of the LCD Ser 369 to Ser 410 engage in the most contacts as in the simulations without explicitly modeling the helix. Phosphorylation of Ser 332 has been shown to disrupt the helical structure (Haider BioPhysJ 2024) and thus one might consider that the simulation residues without the fixed helical structure give a more accurate picture of the contact statistics of the interaction of TDP-43 with the active site of CK1δ. Simulating the coupling between secondary structure and phosphorylation requires a more detailed simulation model (Rizuan JCI 2022, Zhang JACS 2024, Zheng JPCB 2020).”

When discussing the enzymatic phosphorylation of TDP-43 LCD condensate we write “As for simulations in the dilute phase (Supplementary Fig.9), we also checked that we obtained similar results by modeling the helical structure in residues 319-341 explicitly (Supplementary Fig.24, Supplementary Methods). Dissolution of the TDP-43 condensates in simulations with 5 CK1 δ chains, with more and more chains outside the condensate, starts when about 24% of Ser are phosphorylated, Supplementary Fig.24), which is similar to simulations without enforcing the helical structure.”

In addition, we have added a reference to the groundbreaking paper by Dingon et al when describing the re-scaling of interactions of the folded domains to make the origin of this important idea clearer.

That being said, it is worth noting that single-bead-per-residue CG models cannot fully capture the complex nature of site-specific interactions by TDP-43 CR. In particular, these models do not account for how its partial helicity stabilizes and extends across the entire CR upon intermolecular helix–helix interactions, nor do they capture sequence-specific changes to its helical structure that may occur due to phosphorylation of serines within the region, which also modulates its phase separation (Haider et al. 2024, PMID: 38178578).

We agree that single-bead-residue CG models cannot fully capture the modulation of the helical structure in the CR by intermolecular contacts and serine phosphorylation. We write in Discussion: “*More detailed simulation models (Zhang JACS 2024, Rizuan bioRxiv 2024, Zheng JPCB 2020, Rekhi NatChem 2024) could also capture how secondary structure changes upon phosphorylation, as shown for the C-terminal helix of TDP-43 (Haider BioPhysJ 2024).*”

2. An additional point for the authors to consider is the need for a clearer rationale for choosing the HPS-cation- π model.

We chose the HPS-cation- π model as starting point for our simulations based on the paper by Tejedor et al Biophys J 2021 PMID 34762868 and now write in Methods: “... we used the modified HPS model in which cation- π interactions are enhanced (Das PNAS 2020), which was previously shown by Tejedor et al to capture the relative propensity of full-length and LCD TDP-43 to phase separate (Tejedor BioPhysJ 2021)”.

Furthermore, while cation- π interactions are discussed in several instances, it may be beneficial to acknowledge that the drivers of TDP-43 CTD phase separation extend beyond π - π and cation- π interactions. Mohanty et al. (PMID: 37579155) demonstrated that methionine residues, both within and outside the CR, play a crucial role in phase separation. Notably, methionine residues within the CR have a significantly greater impact, suggesting that phase separation is strongly enhanced through helix–helix contacts stabilized by methionine residues in this region. The authors may decide whether and how to incorporate these points, but acknowledging them could provide a more comprehensive perspective on the molecular basis of TDP-43 CTD phase separation.

To better connect our results to the emerging understanding of the molecular basis of TDP-43 CTD phase separation we added to the Introduction: “*Besides the helical structure in the LCD, its enrichment in aromatic and aliphatic residues, including methione, drive its phase separation* (Mohanty PNAS 2023, Schmidt NatComm 2019, Mohanty ProteinSci 2024).”

3. To improve clarity for readers, it may be helpful to include a sequence analysis of CK1 δ , highlighting the surface-exposed residues, active site, and its disordered tail.

We have added a schematic showing the surface-exposed residues, active site, and its disordered tail in the Supplementary Fig.5.

4. The authors mention that when they switch off the arbitrary rescaling of hydrophobic interactions for CK1 δ , the enzyme localizes to the interior of the condensate. Could they comment on how this scenario impacts the results compared to the case where CK1 δ is positioned on the surface of the condensate?

As this is the second round of reviews, I leave it to author's discretion whether to address these points or not.

Understanding how the localization of Ck1 δ in condensates affects phosphorylation rates will be a very exciting extension of this manuscript, which is beyond the scope of the current manuscript. Going forward we can explore how to target the enzyme to the interior of the condensate by modifying its IDR. This could change the phosphorylation rate and how the condensate responds to enzymatic phosphorylation.

Reviewer #3 (Remarks to the Author):

We thank the Reviewer for their important contribution and helping us to improve our manuscript.